# Self-Calibrating Conformal Prediction

**Lars van der Laan**
University of Washington
lvdlaan@uw.edu

**Ahmed M. Alaa**
UC Berkeley and UCSF
amalaa@berkeley.edu

## Abstract

In machine learning, model calibration and predictive inference are essential for producing reliable predictions and quantifying uncertainty to support decision-making. Recognizing the complementary roles of point and interval predictions, we introduce *Self-Calibrating Conformal Prediction*, a method that combines Venn-Abers calibration and conformal prediction to deliver calibrated point predictions alongside prediction intervals with finite-sample validity conditional on these predictions. To achieve this, we extend the original Venn-Abers procedure from binary classification to regression. Our theoretical framework supports analyzing conformal prediction methods that involve calibrating model predictions and subsequently constructing conditionally valid prediction intervals on the same data, where the conditioning set or conformity scores may depend on the calibrated predictions. Real-data experiments show that our method improves interval efficiency through model calibration and offers a practical alternative to feature-conditional validity.

## 1 Introduction

Particularly in safety-critical sectors, such as healthcare, it is important to ensure decisions inferred from machine learning models are reliable, under minimal assumptions [38, 57, 11, 46, 56]. As a response, there is growing interest in predictive inference methods that quantify uncertainty in model predictions via prediction intervals [44, 28]. Conformal prediction (CP) is a popular, model-agnostic, and distribution-free framework for predictive inference, which can be applied post-hoc to any prediction pipeline [59, 49, 3, 34]. Given a prediction issued by a black-box model, CP outputs a prediction interval that is guaranteed to contain the unseen outcome with a user-specified probability [34]. However, a limitation of CP is that this prediction interval only provides valid coverage marginally, when averaged across all possible contexts – with 'context' referring to the information available for decision-making. Constructing informative prediction intervals that offer context-conditional coverage is generally unattainable without making additional distributional assumptions [58, 35, 4]. Consequently, there has been an upsurge in research developing CP methods that offer weaker, yet practically useful, notions of conditional validity; see, e.g., [43, 31, 47, 32, 23, 21].

In prediction settings, *model calibration* is a desirable property of machine learning predictors that ensures that the predicted outcomes accurately reflect the true outcomes [39, 65, 66, 22]. Specifically, a predictor is calibrated for the outcome if the average outcome among individuals with identical predictions is close to their shared prediction value [25]. Such a predictor is more robust against the over-or-under estimation of the outcome in extremes of predicted values. It also has the property that the best prediction of the outcome conditional on the model's prediction is the prediction itself, which facilitates transparent decision-making [51]. There is a rich literature studying post-hoc calibration of prediction algorithms using techniques such as Platt's scaling [45, 14], histogram binning [65, 25, 26], isotonic calibration [66, 40, 51], and Venn-Abers calibration [60].

Given the roles of both point and interval predictions in decision-making, we introduce a dual calibration objective that aims to construct (i) calibrated point predictions and (ii) associated prediction intervals with valid coverage conditional on these point predictions. Marrying model calibration and

predictive inference, we propose a solution to this objective that combines two post-hoc approaches — Venn-Abers calibration [61, 60] and CP [59] — to simultaneously provide point predictions and prediction intervals that achieve our dual objective in finite samples. In doing so, we extend the original Venn-Abers procedure from binary classification to the regression setting. Our theoretical and experimental results support the integration of model calibration into predictive inference methods to improve interval efficiency and interpretability.

## 2 Problem setup

### 2.1 Notation

We consider a standard regression setup in which the input $X \in \mathcal{X} \subset \mathbb{R}^d$ corresponds to contextual information available for decision-making, and the output $Y \in \mathcal{Y} \subset \mathbb{R}$ is an outcome of interest. We assume that we have access to a calibration dataset $\mathcal{C}_n = \{(X_i, Y_i)\}_{i=1}^n$ comprising $n$ *i.i.d.* data points drawn from an unknown distribution $P := P_X P_{Y|X}$. We assume access to a black-box predictor $f : \mathcal{X} \mapsto \mathcal{Y}$, obtained by training an ML model on a dataset that is independent of $\mathcal{C}_n$. Throughout this paper, we do not make any assumptions on the model $f$ or the distribution $P$. For a quantile level $\alpha \in (0, 1)$, we denote the "pinball" quantile loss function $\ell_\alpha$ by $\ell_\alpha(f(x), y) := 1(y \geq f(x)) \cdot \alpha(y - f(x)) + 1(y < f(x)) \cdot (1 - \alpha)(f(x) - y)$.

### 2.2 Conditional predictive inference and a curse of dimensionality

Let $(X_{n+1}, Y_{n+1})$ be a new data point drawn from $P$ independently of the calibration data $\mathcal{C}_n$. **Our high-level aim** is to develop a *predictive inference* algorithm that constructs a prediction interval $\widehat{C}_n(X_{n+1})$ around the point prediction issued by the black-box model, i.e., $f(X_{n+1})$. For this prediction interval to be deemed *valid*, it should *cover* the true outcome $Y_{n+1}$ with a probability $1 - \alpha$. Conformal prediction (CP) is a method for predictive inference that can be applied in a post-hoc fashion to any black-box model [59]. The vanilla CP procedure issues prediction intervals that satisfy the marginal coverage condition:

$$\mathbb{P}(Y_{n+1} \in \widehat{C}_n(X_{n+1})) \geq 1 - \alpha, \tag{1}$$

where the probability $\mathbb{P}$ is taken with respect to the randomness in $\mathcal{C}_n$ and $(X_{n+1}, Y_{n+1})$. However, marginal coverage might lack utility in decision-making scenarios where decisions are context-dependent. A prediction band $\widehat{C}_n(x)$ achieving 95% coverage may exhibit arbitrarily poor coverage for specific contexts $x$. Ideally, we would like this coverage condition to hold for each context $x \in \mathcal{X}$, i.e., the conventional notion of "conditional validity" requires

$$\mathbb{P}(Y_{n+1} \in \widehat{C}_n(X_{n+1}) | X_{n+1} = x) \geq 1 - \alpha, \tag{2}$$

for all $x \in \mathcal{X}$. However, previous work has shown that it is impossible to achieve (2) without distributional assumptions [58, 35, 4].

While context-conditional validity as in (2) is generally unachievable, it is feasible to attain weaker forms of conditional validity. Given any finite set of groups $\mathcal{G}$ and a grouping function $G : \mathcal{X} \times \mathcal{Y} \to \mathcal{G}$, Mondrian-CP offers coverage conditional on group membership, that is, $\mathbb{P}(Y_{n+1} \in \widehat{C}_n(X_{n+1}) | G(X_{n+1}, Y_{n+1}) = g) \geq 1 - \alpha, \ \forall g \in \mathcal{G}$ [59, 47]. Expanding upon group- and context-conditional coverage, A *multicalibration* objective was introduced in [17] that seeks to satisfy, for all $h$ in an (infinite-dimensional) class $\mathcal{F}$ of weighting functions (i.e., 'covariate shifts'), the property:

$$\mathbb{E}\big[h(X_{n+1})\big\{(1 - \alpha) - 1\{Y_{n+1} \in \widehat{C}_n(X_{n+1})\}\big\}\big] = 0. \tag{3}$$

Gibbs et al. [21] proposed a regularized CP framework for (approximately) achieving (3) that provides a means to trade off the efficiency (i.e., width) of prediction intervals and the degree of conditional coverage achieved. However, Barber et al. [4] and Gibbs et al. [21] establish the existence of a "curse of dimensionality": as the dimension of the context increases, smaller classes of weighting functions must be considered to retain the same level of efficiency. For group-conditional coverage, this curse of dimensionality manifests in the size of the subgroup class $\mathcal{G}$ [4] via its VC dimension [55]. Thus, especially in data-rich contexts, prediction intervals with meaningful multicalibration guarantees over the context space may be too wide for decision-making.

## 2.3 A dual calibration objective

In decision-making, both point predictions and prediction intervals play a role. For example, in scenarios with a low signal-to-noise ratio, prediction intervals may be too wide to directly inform decision-making, as their width is typically of the order of the standard deviation of the outcome. Point predictions might be used to guide decisions, while prediction intervals help quantify deviations of point predictions from unseen outcomes and assess the risk associated with these decisions.

Viewing the black-box model $f$ as a scalar dimension reduction of the context $x$, a natural relaxation of the infeasible objective of context-conditional validity in (2) is prediction-conditional validity, i.e., $P(Y_{n+1} \in \widehat{C}_n(X_{n+1})|f(X_{n+1})) \geq 1 - \alpha$. Prediction-conditional validity ensures that the interval widths adapt to the outputs of the model $f(\cdot)$, so that the intervals can be reliably used to quantify the deviation of model predictions from unseen outcomes. Since prediction-conditional validity only requires coverage conditional on a one-dimensional random variable, it avoids the curse of dimensionality associated with context-conditional validity. In addition, as illustrated in our experiments in Section 5 and Appendix C, when the heteroscedasticity (e.g., variance) in the outcome is a function of its conditional mean, prediction-conditional validity can closely approximate context-conditional validity, so long as the predictor estimates the conditional mean of the outcome sufficiently well.

Given the roles of both point and interval predictions in decision-making, we introduce a novel dual calibration objective, *self-calibration*, that aims to construct (i) calibrated point predictions and (ii) associated prediction intervals with valid coverage conditional on these point predictions. Formally, given the model $f$ and calibration data $\mathcal{C}_n \cup \{X_{n+1}\}$, **our objective** is to post-hoc construct a calibrated point prediction $f_{n+1}(X_{n+1})$ and a compatible prediction interval $\widehat{C}_{n+1}(X_{n+1})$ centered around $f_{n+1}(X_{n+1})$ that satisfies the following desiderata:

> (i) **Perfectly Calibrated Point Prediction:** $f_{n+1}(X_{n+1}) = \mathbb{E}[Y_{n+1} \mid f_{n+1}(X_{n+1})]$.
>
> (ii) **Prediction-Conditional Validity:** $\mathbb{P}(Y_{n+1} \in \widehat{C}_{n+1}(X_{n+1})|f_{n+1}(X_{n+1})) \geq 1 - \alpha$.

Desideratum (i) states that the point prediction $f_{n+1}(X_{n+1})$ should be *perfectly calibrated* — or self-consistent [19] — for the true outcome $Y_{n+1}$ [37, 25, 51]. It is widely recognized that the calibration of model predictions is important to ensure their reliability, trustworthiness, and interpretability in decision-making [37, 65, 7, 24, 15]. Desideratum (i) also improves interval efficiency by ensuring $\widehat{C}_{n+1}(X_{n+1})$ is centered around an unbiased prediction, meaning the interval's width is driven by outcome variation rather than by prediction bias. Desideratum (ii) is a prediction interval variant of (i) that ensures the prediction interval $\widehat{C}_{n+1}(X_{n+1})$ is calibrated with respect to the model $f_{n+1}$, providing valid coverage for $Y_{n+1}$ within contexts with the same *calibrated* point prediction. We refer to a predictive inference algorithm simultaneously satisfying (i) and (ii) as *self-calibrating*, as such a procedure is automatically able to adapt to miscalibration in the model $f(\cdot)$ due to, e.g., model misspecification or distribution shifts, ensuring that the interval is constructed from a calibrated predictive model.

Self-calibration can also be motivated by a decision-making scenario where point predictions determine actions and prediction intervals are used to apply these actions selectively. When point predictions are sufficient statistics for actions, self-calibration implies that the point and interval predictions are accurate, on average, within the subset of all contexts receiving the same prediction and, therefore, the same action.

## 3 Self-Calibrating Conformal Prediction

A key advantage of CP is that it can be applied post-hoc to any black-box model $f$ without disrupting its point predictions. However, desideratum (i) introduces a perfect calibration requirement for the point predictions of $f$, thereby interfering with the underlying model specification. In this section, we introduce Self-Calibrating CP (SC-CP), a modified version of CP that is self-calibrating in that it satisfies (i) and (ii), while preserving all the favorable properties of CP, including its finite-sample validity and post-hoc applicability. Before describing our complete procedure in Section 3.3, we provide background on point calibration and propose Venn-Abers calibration for regression.

## 3.1 Preliminaries on point calibration

Following the framing of van der Laan et al. [51] (see also [25]), a *point calibrator* is a post-hoc procedure that aims to learn a transformation $\theta_n : \mathbb{R} \to \mathbb{R}$ of the black-box model $f$ such that: (1) $\theta_n(f(X_{n+1}))$ is well-calibrated for $Y_{n+1}$ in the sense of Desideratum (i); and (2) $\theta_n \circ f$ is comparably predictive to $f$. Condition (2) ensures that in the process of achieving (1), the quality of the model $f$ is not compromised, and excludes trivial calibrators such as $\theta_n(f(\cdot)) := \frac{1}{n}\sum_{i=1}^{n} Y_i$ [26]. To our knowledge, this notion of calibration traces back to Mincer and Zarnowitz (1969) [39], who introduced the idea of regressing outcomes on predictions to achieve calibration in forecasting.

Commonly-employed point calibrators include Platt's scaling [45, 14], histogram (or quantile) binning [65], and isotonic calibration [66, 40]. Mechanistically, these point calibrators learn $\theta_n$ by regressing the outcomes $\{Y_i\}_{i=1}^{n}$ on the model predictions $\{f(X_i)\}_{i=1}^{n}$. Importantly, however, point calibration fundamentally differs from the regression task of learning $E_P[Y|f(X)]$, as calibration can be achieved without smoothness assumptions, allowing for misspecification of the regression task [25]. Histogram binning is a simple and distribution-free calibration procedure [25, 26] that learns $\theta_n$ via a histogram regression over a finite (outcome-agnostic) binning of the output space $f(\mathcal{X})$. Isotonic calibration is an outcome-adaptive binning method that uses isotonic regression [5] to learn $\theta_n$ by minimizing the empirical mean square error over all 1D piece-wise constant, monotone nondecreasing transformations. Isotonic calibration is distribution-free — it does not rely on monotonicity assumptions — and, in contrast with histogram binning, it is tuning parameter-free and naturally preserves the mean-square error of the original predictor (as the identity transform is monotonic) [51]. A key limitation of histogram binning and isotonic calibration is that their calibration guarantees are only approximate, and desideratum (i) only holds asymptotically.

## 3.2 Venn-Abers calibration

For binary classification, Vovk and Petej [60] proposed Venn-Abers calibration, which iterates isotonic calibration over imputations $y \in \mathcal{Y}$ of the unseen outcome $Y_{n+1}$ to provide calibrated multi-probabilistic predictions in finite samples. In this section, we generalize the Venn-Abers calibration procedure to regression, offering finite-sample calibration guarantees for non-binary outcomes.

Let $\Theta_{\text{iso}}$ consist of all univariate, piecewise constant functions that are monotonically nondecreasing. Our Venn-Abers calibration procedure, outlined in Alg. 1, is derived from an *oracle* variant of isotonic calibration that provides a perfectly calibrated point prediction *in finite samples*, but requires knowledge of the true outcome $Y_{n+1}$. Specifically, the Venn-Abers calibration algorithm iterates over imputed outcomes $y \in \mathcal{Y}$ for $Y_{n+1}$ and applies isotonic calibration to the augmented dataset $\mathcal{C}_n \cup \{(X_{n+1}, y)\}$ to produce a set of point predictions $f_{n,X_{n+1}}(X_{n+1}) := \{f_n^{(X_{n+1},y)}(X_{n+1}) : y \in \mathcal{Y}\}\}$. When the outcome space $\mathcal{Y}$ is non-discrete, Alg. 1 may be infeasible to compute exactly and can be approximated by discretizing $\mathcal{Y}$. Nonetheless, the range of the Venn-Abers multi-prediction can be feasibly computed as $[f_n^{(x,y_{\min})}(x), f_n^{(x,y_{\max})}(x)]$ where $[y_{\min}, y_{\max}] := \text{range}(\mathcal{Y})$, in light of the min-max representation of isotonic regression [33].

Unlike point calibrators, Venn-Abers calibration generates a set of calibrated predictions for each context $X_{n+1}$, indexed by $y \in \mathcal{Y}$. As we demonstrate later, this set prediction is guaranteed *in finite samples* to include a perfectly calibrated point prediction, namely, the oracle prediction $f_n^{(X_{n+1},Y_{n+1})}(X_{n+1})$ corresponding to the true outcome $Y_{n+1}$. Moreover, each prediction in the set, being obtained via isotonic calibration, still enjoys the same large-sample calibration guarantees as isotonic calibration [51]. By the stability of isotonic regression, as the size of the calibration set $n$ increases, the width of this set of predictions rapidly narrows, eventually converging to a single, perfectly calibrated point prediction [60]. Venn-Abers calibration thus provides a measure of epistemic uncertainty by producing a range of values for a perfectly calibrated point prediction. In cases with small sample sizes, standard isotonic calibration can overfit, leading to poorly calibrated point predictions. When this overfitting occurs, the Venn-Abers set prediction widens, reflecting greater uncertainty in the perfectly calibrated point prediction within the set [30].

For the binary classification case, [60] derived a (large-sample) calibrated point prediction from the Venn-Abers multi-prediction using a shrinkage approach. We can similarly construct, for each $x \in \mathcal{X}$,

**Algorithm 1** Venn-Abers Calibration

---

**Input:** Calibration data $\mathcal{C}_n = \{(X_i, Y_i)\}_{i=1}^n$, model $f$, context $x \in \mathcal{X}$.

1: **for** each $y \in \mathcal{Y}$ **do**
2:     Set augmented dataset $\mathcal{C}_n^{(x,y)} := \mathcal{C}_n \cup \{(x,y)\}$;
3:     Apply isotonic calibration to $f$ using $\mathcal{C}_n^{(x,y)}$:

$$\theta_n^{(x,y)} := \operatorname{argmin}_{\theta \in \Theta_{\text{iso}}} \sum_{i \in \mathcal{C}_n^{(x,y)}} \{Y_i - \theta \circ f(X_i)\}^2.$$
$$f_n^{(x,y)} := \theta_n^{(x,y)} \circ f.$$

4: **end for**
**Output:** Multi-prediction $\{f_n^{(x,y)}(x) : y \in \mathcal{Y}\}$.

---

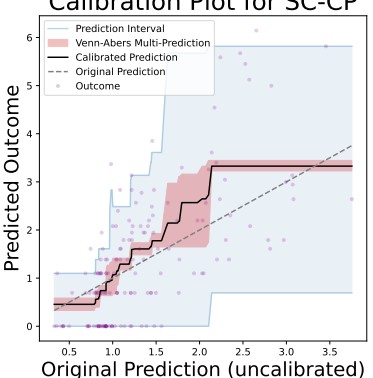

Figure 1: Example SC-CP output with small $\mathcal{C}_n$ ($n = 200$).

**Algorithm 2** Self-Calibrating Conformal Prediction

---

**Input:** Calibration data $\mathcal{C}_n = \{(X_i, Y_i)\}_{i=1}^n$, model $f$, context $x \in \mathcal{X}$, miscoverage level $\alpha \in (0,1)$

1: **for** each $y \in \mathcal{Y}$ **do**
2:     Obtain calibrated model $f_n^{(x,y)}$ by isotonic calibrating $f$ on $\mathcal{C}_n \cup \{(x,y)\}$ as in Alg. 1;
3:     Set self-calibrating conformity scores $S_i^{(x,y)} = |Y_i - f_n^{(x,y)}(X_i)|, \forall i \in [n]$ and $S_{n+1}^{(x,y)} = |y - f_n^{(x,y)}(x)|$;
4:     Calculate $1 - \alpha$ empirical quantile $\rho_n^{(x,y)}(x)$ of conformity scores with same calibrated prediction as $x$:

$$\operatorname*{argmin}_{q \in \mathbb{R}} \sum_{i=1}^n \mathbf{1}\Big\{ f_n^{(x,y)}(X_i) = f_n^{(x,y)}(x) \Big\} \cdot \ell_\alpha(q, S_i^{(x,y)}) + \ell_\alpha(q, S_{n+1}^{(x,y)});$$

5: **end for**
6: Set $f_{n+1}(x) := \{f_n^{(x,y)}(x) : y \in \mathcal{Y}\}$.
7: Set $\widehat{C}_{n+1}(x) := \{y \in \mathcal{Y} : |y - f_n^{(x,y)}(x)| \leq \rho_n^{(x,y)}(x)]\}$.
**Output:** $f_{n+1}(x) \subset \text{conv}(\mathcal{Y}), \widehat{C}_{n+1}(x) \subset \mathcal{Y}$

---

a point prediction as follows:

$$\widetilde{f}_{n+1,x}(x) := f_{n+1,x}^{\text{mid}}(x) + \frac{f_n^{(x,y_{\max})}(x) - f_n^{(x,y_{\min})}(x)}{y_{\max} - y_{\min}} \left\{ \overline{y}_n - f_{n+1,x}^{\text{mid}}(x) \right\}, \tag{4}$$

where $f_{n+1,x}^{\text{mid}}(x) := \frac{1}{2}\{f_n^{(x,y_{\max})}(x) + f_n^{(x,y_{\min})}(x)\}$ is the midpoint of the multi-prediction and $\overline{y}_n := \frac{1}{n} \sum_{i=1}^n Y_i$. The behavior of $\widetilde{f}_{n+1,x}(x)$ is natural; it shrinks the point prediction $f_{n+1,x}^{\text{mid}}(x)$ towards the average outcome $\overline{y}_n$ (a well-calibrated prediction) proportional to how uncertain we are in the calibration of $f_{n+1,x}^{\text{mid}}(x)$. The ratio $\frac{1}{y_{\max}-y_{\min}}(f_n^{(x,y_{\max})}(x) - f_n^{(x,y_{\min})}(x))$ measures the sensitivity of isotonic regression to the addition of a single data point to $\mathcal{C}_n$, and a value closer to 1 corresponds to a higher degree of overfitting. In the extreme case where the calibration dataset is very large, we have $f_n^{(x,y_{\max})}(x) \approx f_n^{(x,y_{\min})}(x)$, implying that $\widetilde{f}_{n+1,x}(x) \approx f_{n+1,x}^{\text{mid}}(x)$. Conversely, in the opposite extreme where the calibration dataset is very small and isotonic regression overfits, we have $f_n^{(x,y_{\max})}(x) \approx y_{\max}$ and $f_n^{(x,y_{\min})}(x) \approx y_{\min}$, in which case $\widetilde{f}_{n+1,x}(x) \approx \overline{y}_n$. We could replace $\overline{y}_n$ in (4) with any reference predictor, such as one calibrated using Platt's scaling or quantile-binning.

### 3.3 Conformalizing Venn-Abers Calibration

In this section, we propose Self-Calibrating Conformal Prediction, which conformalizes the Venn-Abers calibration procedure to provide prediction intervals centered around the Venn-Abers multi-prediction that are self-calibrated in the sense of desiderata (i) and (ii).

A simple, albeit naive, strategy for achieving (i) and (ii) without finite-sample guarantees involves using the dataset $\mathcal{C}_n$ to calibrate point predictions of $f(\cdot)$ through isotonic regression, and then constructing prediction intervals from the $1 - \alpha$ empirical quantiles of prediction errors within subgroups defined by unique values of the calibrated point predictions. To motivate our SC-CP algorithm, we introduce an infeasible variant of this seemingly naive procedure that is valid in finite samples, but can only be computed by an oracle that knows the unseen outcome $Y_{n+1}$. In

this oracle procedure, we compute a perfectly calibrated prediction $f^*_{n+1}(X_{n+1}) := \theta^*_{n+1}(f(X_{n+1}))$ by isotonic calibrating $f$ using the oracle-augmented calibration set $\{(X_i, Y_i)\}^{n+1}_{i=1}$, where $\theta^*_{n+1} \in \arg\min_{\theta \in \Theta_{\text{iso}}} \sum^{n+1}_{i=1} \{Y_i - \theta(f(X_i))\}^2$. Next, we compute the conformity scores $S^*_i := |Y_i - f^*_{n+1}(X_i)|$ as the absolute residuals of the calibrated predictions. An oracle prediction interval is then given by $C^*_{n+1}(X_{n+1}) := f^*_{n+1}(X_{n+1}) \pm \rho^*_{n+1}(X_{n+1})$, where $\rho^*_{n+1}(X_{n+1})$ is the empirical $1 - \alpha$ quantile of conformity scores with calibrated predictions identical to $X_{n+1}$, that is, scores in the set $\{S^*_i : f^*_{n+1}(X_i) = f^*_{n+1}(X_{n+1}), i \in [n+1]\}$. Importantly, isotonic regression, which is an outcome-adaptive histogram binning method, ensures that the calibrated model $f^*_{n+1}$ is piece-wise constant, with a sufficiently large number of observations averaged within each constant segment—typically on the order of $n^{2/3}$ [16]. Consequently, the empirical quantile $\rho^*_{n+1}(X_{n+1})$ is generally stable with relatively low variability across realizations of $\mathcal{C}_n$. In our proofs, using the first-order conditions characterizing the optimizer $\theta_{n+1}$ and exchangeability, we show that $f^*_{n+1}(X_{n+1}) = \mathbb{E}[Y_{n+1} | f^*_{n+1}(X_{n+1})]$, so that desideratum (i) is satisfied. Furthermore, we establish that the interval $C^*_{n+1}(X_{n+1})$ achieves desideratum (ii), i.e., $\mathbb{P}(Y_{n+1} \in C^*_{n+1}(X_{n+1}) \mid f^*_{n+1}(X_{n+1})) \geq 1 - \alpha$. To do so, our key insight is that $\rho^*_{n+1}(X_{n+1})$ corresponds to the evaluation of the function $\rho^*_{n+1}$ computed via prediction-conditional quantile regression as:

$$\rho^*_{n+1} \in \operatorname*{argmin}_{\theta \circ f^*_{n+1}; \theta : \mathbb{R} \to \mathbb{R}} \frac{1}{n+1} \sum^{n+1}_{i=1} \ell_\alpha \left( \theta \circ f^*_{n+1}(X_i), S_i \right).$$

The first-order conditions characterizing the optimizer $\rho^*_{n+1}$ combined with the exchangeability between $\mathcal{C}_n$ and $(X_{n+1}, Y_{n+1})$ can be used to show that $C^*_{n+1}(X_{n+1})$ is *multi-calibrated* against the class of weighting functions $\mathcal{F}_{n+1} := \{\theta \circ f^*_{n+1}; \theta : \mathbb{R} \to \mathbb{R}\}$ in the sense of (3). Using first-order conditions to establish the theoretical validity of conformal prediction was also applied by Gibbs et al. [21] to demonstrate the multi-calibration of oracle prediction intervals obtained from quantile regression over a fixed class $\mathcal{F}$. In our case, quantile regression is performed over a data-dependent function class $\mathcal{F}_{n+1}$, learned from the calibration data, which introduces additional challenges in our proofs.

Our SC-CP method, which is outlined in Alg. 2, follows a similar procedure to the above oracle procedure. Since the new outcome $Y_{n+1}$ is unobserved, we instead iterate the oracle procedure over all possible imputed values $y \in \mathcal{Y}$ for $Y_{n+1}$. As in Alg. 1., this yields a set of isotonic calibrated models $f_{n,X_{n+1}} := \{f^{(X_{n+1},y)}_n : y \in \mathcal{Y}\}$, where $f_{n+1}(X_{n+1})$ is the Venn-Abers multi-prediction of $Y_{n+1}$. Then, for each $y \in \mathcal{Y}$ and $i \in [n]$, we define the *self-calibrating conformity scores* $S^{(X_{n+1},y)}_i := |Y_i - f^{(X_{n+1},y)}_n(X_i)|$ and $S^{(X_{n+1},y)}_{n+1} := |y - f^{(X_{n+1},y)}_n(X_{n+1})|$, where the dependency of our scores on the imputed outcome $y \in \mathcal{Y}$ is akin to Full (or transductive) CP [59]. Our SC-CP interval is then given by $\widehat{C}_{n+1}(X_{n+1}) := \{y \in \mathcal{Y} : S^{(X_{n+1},y)}_{n+1} \leq \rho^{(X_{n+1},y)}_n(X_{n+1})\}$, where $\rho^{(X_{n+1},y)}_n(X_{n+1})$ is the empirical $1 - \alpha$ quantile of the level set $\{S^{(X_{n+1},y)}_i : f^{(X_{n+1},y)}_n(X_i) = f^{(X_{n+1},y)}_n(X_{n+1}), i \in [n+1]\}$. By definition, $\widehat{C}_{n+1}(X_{n+1})$ covers $Y_{n+1}$ if, and only if, the oracle interval $C^*_{n+1}(X_{n+1})$ covers $Y_{n+1}$, thereby inheriting the self-calibration of $C^*_{n+1}(X_{n+1})$. Formally, $\widehat{C}_{n+1}(X_{n+1})$ is a set, but it can be converted to an interval by taking its range, with little efficiency loss.

### 3.4 Computational considerations

The main computational cost of Alg. 1 and Alg. 2 is in the isotonic calibration step, executed for each $y \in \mathcal{Y}$. Isotonic regression [5] can be scalably and efficiently computed using implementations of `xgboost` [12] for univariate regression trees with monotonicity constraints. Similar to Full (or transductive) CP [59], Alg. 2 may be computationally infeasible for non-discrete outcomes, and can be approximated by iterating over a finite subset of $\mathcal{Y}$. In our implementation, we iterate over a grid of $\mathcal{Y}$ and use linear interpolation to impute the threshold $\rho^{(x,y)}_n(x)$ and score $S^{(x,y)}_{n+1}$ for each $y \in \mathcal{Y}$. As with Full CP and multicalibrated CP [21], Alg. 1 and Alg. 2 must be separately applied for each context $x \in \mathcal{X}$. The algorithms depend solely on $x \in \mathcal{X}$ through its prediction $f(x)$, so we can approximate the outputs for all $x \in \mathcal{X}$ by running each algorithm for a finite number of $x \in \mathcal{X}$ corresponding to a finite grid of the 1D output space $f(\mathcal{X}) = \{f(x) : x \in \mathcal{X}\} \subset \mathbb{R}$. In addition, both algorithms are fully parallelizable across both the input context $x \in \mathcal{X}$ and the imputed outcome $y \in \mathcal{Y}$. In our implementation, we use nearest neighbor interpolation in the prediction space to impute outputs for each $x \in \mathcal{X}$. In our experiments with sample sizes ranging from $n = 5000$

to 40000, quantile binning of both $f(\mathcal{X})$ and $\mathcal{Y}$ into 200 equal-frequency bins enables execution of Alg. 1 and Alg. 2 across all contexts in minutes with negligible approximation error.

# 4 Theoretical guarantees

In this section, under exchangeability of the data, we establish that the Venn-Abers multi-prediction $f_{n,X_{n+1}}(X_{n+1}) := \{f_n^{(X_{n+1},y)}(X_{n+1}) : y \in \mathcal{Y}\}$ and SC-CP interval $\widehat{C}_{n+1}(X_{n+1})$ output by Alg. 2 satisfy desiderata (i) and (ii) in finite samples and without distributional assumptions. Under an *iid* condition, we further establish that, asymptotically, the Venn-Abers calibration step within the SC-CP algorithm results in better point predictions and, consequently, more efficient prediction intervals.

The following theorem establishes that the Venn-Abers multi-prediction is perfectly calibrated in the sense of [61], containing a perfectly calibrated point prediction of $Y_{n+1}$ in finite samples.

C1) *Exchangeability:* $\{(X_i, Y_i)\}_{i=1}^{n+1}$ are exchangeable.

C2) *Finite second moment:* $E_P[Y^2] < \infty$.

**Theorem 4.1** (Perfect calibration of Venn-Abers multi-prediction). *Under Conditions C1 and C2, the Venn-Abers multi-prediction $f_{n,X_{n+1}}(X_{n+1})$ almost surely satisfies the condition $f_n^{(X_{n+1},Y_{n+1})}(X_{n+1}) = \mathbb{E}[Y_{n+1} \mid f_n^{(X_{n+1},Y_{n+1})}(X_{n+1})]$.*

Theorem 4.1 generalizes an analogous result by [60] for the special case of binary classification. Even in this special case, our proof is novel and elucidates how Venn-Abers calibration uses exchangeability with the least-squares loss in a manner analogous to how CP uses exchangeability with the quantile loss [21].

The following theorem establishes desideratum (ii) for the interval $\widehat{C}_{n+1}(X_{n+1})$ with respect to the oracle prediction $f_n^{(X_{n+1},Y_{n+1})}(X_{n+1})$ of Theorem 4.1. In what follows, let $\mathrm{polylog}(n)$ be a given sequence that grows polynomially logarithmically in $n$.

C3) The conformity scores $|Y_i - f_n^{(X_{n+1},Y_{n+1})}(X_i)|, \forall i \in [n+1]$, are almost surely distinct.

C4) The number of constant segments for $f_n^{(X_{n+1},Y_{n+1})}$ is at most $n^{1/3} \mathrm{polylog}(n)$.

**Theorem 4.2** (Self-calibration of prediction interval). *Under C1, it holds almost surely that*

$$\mathbb{P}\left(Y_{n+1} \in \widehat{C}_{n+1}(X_{n+1}) \mid f_n^{(X_{n+1},Y_{n+1})}(X_{n+1})\right) \geq 1 - \alpha.$$

*If also C3 and C4 hold, then* $\mathbb{E}\left|\alpha - \mathbb{P}\left(Y_{n+1} \notin \widehat{C}_{n+1}(X_{n+1}) \mid f_n^{(X_{n+1},Y_{n+1})}(X_{n+1})\right)\right| \leq \frac{\mathrm{polylog}(n)}{n^{2/3}}.$

Theorem 4.2 says that $\widehat{C}_{n+1}(X_{n+1})$ satisfies desideratum (ii) with coverage that is, on average, nearly exact up to a factor $\frac{\mathrm{polylog}(n)}{n^{2/3}}$. Notably, the deviation error from exact coverage tends to zero at a fast dimensionless rate and, therefore, does not suffer from a "curse of dimensionality". Condition C3 is only required to establish the upper coverage bound and is standard in CP - see, e.g., [36, 21]. Although it may fail for non-continuous outcomes, this condition can be avoided by adding a small amount of noise to all outcomes [36]. The constant segment number of $n^{1/3} \mathrm{polylog}(n)$ in C4 is motivated by the theoretical properties of isotonic regression; Assuming C2 and continuous differentiability of $t \mapsto E_P[Y \mid f(X) = t]$, it is shown in [16] that the number of observations in a given constant segment of an isotonic regression solution concentrates in probability around $n^{2/3}$. In general, without C4, our proof establishes a miscoverage upper bound of $\frac{1}{n+1}\mathbb{E}[N_{n+1}]$, where $\mathbb{E}[N_{n+1}]$ is the expected number of constant segments of $f_n^{(X_{n+1},Y_{n+1})}$.

The next theorem examines the interaction between calibration and CP within SC-CP in terms of efficiency of the self-calibrating conformity scores. In the following, let $x \in \mathcal{X}, y \in \mathcal{Y}$ be arbitrary. For each $\theta \in \Theta_{iso}$, define the $\theta$-transformed conformity scoring function $S_\theta : (x', y') \mapsto |y' - \theta \circ f(x')|$. Let $\theta_0 := \mathrm{argmin}_{\theta \in \Theta_{iso}} \int \{S_\theta(x', y')\}^2 dP(x', y')$ be the optimal isotonic transformation of $f(\cdot)$ that minimizes the population mean-square error. Define the self-calibrating conformity scoring function as $S_n^{(x,y)}(x', y') := |y' - f_n^{(x,y)}(x')|$, where $f_n^{(x,y)}$ is obtained as in Alg. 1.

**C5)** *Independent data:* $\{(X_i, Y_i)\}_{i=1}^{n+1}$ are *iid*.

**C6)** *Bounded outcomes:* $\mathcal{Y}$ is a uniformly bounded set.

**Theorem 4.3.** *Under C5 and C6, we have* $\int \{S_n^{(x,y)}(x', y') - S_{\theta_0}(x', y')\}^2 dP(x', y') = O_p(n^{-2/3})$.

The above theorem indicates that the self-calibrating scoring function $S_n^{(x,y)}$ used in Alg. 2 asymptotically converges in mean-square error to the oracle scoring function $S_{\theta_0}$ at a rate of $n^{-2/3}$. Since the oracle scoring function $S_{\theta_0}$ corresponds to a model $\theta_0 \circ f$ with better mean square error than $f$, we heuristically expect that the Venn-Abers scoring function $S_n^{(x,y)}$ will translate to narrower CP intervals, at least asymptotically. We provide experimental evidence for this heuristic in Section 5.

**Limitations.** The perfectly calibrated prediction $f_n^{(X_{n+1}, Y_{n+1})}(X_{n+1})$, guaranteed to lie by Theorem 4.1 in the Venn-Abers multi-prediction, typically cannot be determined precisely without knowledge of $Y_{n+1}$. However, the stability of isotonic regression implies that the width of multi-prediction $f_{n+1}(X_{n+1})$ shrinks towards zero very quickly as the size of the calibration set increases [10]. Moreover, the large-sample theory for isotonic calibration in [51] demonstrates that the $\ell^2$-calibration error of each model $f_n^{(X_{n+1}, y)}$ with $y \in \mathcal{Y}$ is $O_p(n^{-2/3})$. One caveat of SC-CP intervals is that desideratum (ii) is satisfied with respect to the unknown, oracle point prediction $f_n^{(X_{n+1}, Y_{n+1})}(X_{n+1})$. However, we know that this oracle prediction lies within the Venn-Abers multi-prediction by Theorem 4.1, and its value can be determined with high precision with relatively small calibration sets [60] — see, e.g., Figure 1. These limitations appear to be unavoidable as perfectly calibrated point predictions can generally not be constructed in finite samples without oracle knowledge [61, 60].

## 4.1 Related work

The work of [42] proposes a regression extension of Venn-Abers calibration that differs from ours, both algorithmically and in its objective. While our extension constructs a calibrated point prediction $f(X)$ of $Y$ such that $f(X) = \mathbb{E}[Y \mid f(X)]$, their approach uses the original Venn-Abers calibration procedure of [60] to construct a distributional prediction $f_t(X)$ of $1(Y \le t)$ that satisfies $f_t(X) = \mathbb{P}(Y \le t \mid f_t(X))$ for $t \in \mathcal{Y}$.

The impossibility results of [25] imply that any universal procedure providing prediction-conditionally calibrated intervals must explicitly or implicitly discretize the output of the model $f(\cdot)$. The works of [31] and [29] apply Mondrian CP [59] within leaves of a regression tree $f$ to construct prediction intervals with prediction-conditional validity. However, this approach is restricted to tree-based predictors and does not guarantee calibrated point predictions and self-calibrated intervals. Mondrian conformal predictive distributions were applied within bins of model predictions in [9] to satisfy a coarser, distributional form of prediction-conditional validity. A limitation of Mondrian-CP approaches to prediction-conditional validity is that they require pre-specification of a binning scheme for the predictor $f(\cdot)$, which introduces a trade-off between model performance and the width of prediction intervals, and they do not perform point calibration (desideratum (i)) and, thereby, do not guarantee self-calibration. In contrast, SC-CP data-adaptively discretizes the predictor $f(\cdot)$ using isotonic calibration and, in doing so, provides calibrated predictions, improved conformity scores, and self-calibrated intervals.

Other notions of conditional validity have been proposed that, like prediction-conditional validity and self-calibration, avoid the curse of dimensionality of context-conditional validity. In the multiclassification setup, [50] and [18] use Mondrian CP to provide prediction intervals with valid coverage conditional on the class label (i.e., outcome). In [8], Mondrian CP is applied within bins categorized by context-specific difficulty estimates, such as conditional variance estimates. Multivalid-CP [32, 6] offers coverage based on a threshold defining the prediction interval. For multiclassification, [41] propose a procedure for attaining valid coverage conditional on the prediction set size [2].

# 5 Real-Data Experiments: predicting utilization of medical services

## 5.1 Experimental setup

In this experiment, we illustrate how prediction-conditional validity can approximate context-conditional validity when the heteroscedasticity in outcomes is strongly associated with model

predictions, thereby ensuring validity across critical subgroups without their pre-specification. We analyze the Medical Expenditure Panel Survey (MEPS) dataset [1], supplied by the Agency for Healthcare Research and Quality [13], which was used in [47] for Mondrian CP with fairness applications. We use the preprocessed dataset acquired using the Python package cqr, also associated with [47]. This dataset contains $n = 15,656$ observations and $d = 139$ features, and includes information such as age, marital status, race, and poverty status, alongside medical service utilization. Our objective is to predict each individual's healthcare system utilization, represented by a score that reflects visits to doctors' offices, hospital visits, etc. Following [47], we designate race as the sensitive attribute $A$, aiming for *equalized coverage*, where $A = 0$ represents non-white individuals ($n_0 = 9640$) and $A = 1$ represents white individuals ($n_1 = 6016$). The outcome variable $Y$ is transformed by $Y = \log(1 + \text{utilization score})$ to address the skewness of the raw score. In Appendix B, we present additional experimental results for the *Concrete*, *Community*, *STAR*, *Bike*, and *Bio* datasets used in [48] and publicly available in the Python package cqr, associated with [48] and [47].

We randomly partition the dataset into three segments: a training set (50%) for model training, a calibration set (30%) for CP, and a test set (20%) for evaluation. For training the initial model $f(\cdot)$, we use the xgboost [12] implementation of gradient boosted regression trees [20], where maximum tree depth, boosting rounds, and learning rate are tuned using 5-fold cross-validation. We consider two settings for training the model. In **Setting A**, we train the initial model on the untransformed outcomes and then transform the predictions as $\hat{y} \mapsto \log(1 + \hat{y})$, which makes the model predictive but poorly calibrated because it overestimates the true outcomes, in light of Jensen's inequality. In **Setting B**, we train the initial model on the transformed outcomes, leading to fairly well-calibrated predictions. In both settings, calibration and evaluation are applied to the transformed outcomes.

For direct comparison, we compare **SC-CP** with baselines that leverage the standard absolute residual scoring function $S(x, y) := |y - f(x)|$ and target either marginal validity or prediction-conditional validity. The baselines are: **Marginal** CP [34], **Mondrian** CP with categories defined by bins of model predictions [59, 9], **CQR** [48] with model predictions used as features, and the **Kernel**-smoothed conditional CP approach of [21] with model predictions $\{f(X_i)\}_{i=1}^{n}$ used as features and bandwidth tuned with cross-validation. Due to the slow computing time of the implementation provided by [21], we apply **Kernel** on a subset of the calibration data of size $n_{cal} = 500$. **SC-CP** is implemented as described in Alg. 2, using isotonic regression constrained to have at least 20 observations averaged within each constant segment to mitigate overfitting (via the minimum child weight argument of xgboost). The miscoverage level is taken to be $\alpha = 0.1$. **SC-CP** provides calibrated point predictions and self-calibrated intervals, while the **Mondrian** and **Kernel** baselines offer approximate prediction-conditional validity, and **Marginal** and **CQR** only guarantee marginal coverage. We report empirical coverage, average interval width, and calibration error of model predictions in the test set within the sensitive attribute. Calibration error is defined as the mean error of the point predictions, $\mathbb{E}[\hat{Y} - Y \mid A]$ within the sensitive attribute $A$, which measures model over- or under-confidence. For **SC-CP**, we use the calibrated point predictions from (4), while the original point predictions are used for **Marginal**, **Mondrian**, and **Kernel**. For **CQR**, we use an estimate of the conditional median, obtained from a separate xgboost quantile regression model, as the point prediction. We note that, since quantiles are preserved under monotone transformations of the outcome, we expect the conditional median model of **CQR** to be well-calibrated, at least in a median sense, in both **Setting A** and **Setting B**. We include the baseline **Mondrian*** for direct comparison with SC-CP, in which **Mondrian** is applied with the same number of prediction bins as data-adaptively selected by SC-CP.

## 5.2 Results and discussion

The experimental results for each setting are depicted in Figure 2. Each panel's left-most plot showcases a calibration plot [62] for **SC-CP**, illustrating original and calibrated predictions alongside prediction bands. On the right, the panels display prediction bands of our baselines as a function of the original model predictions. Visually, as expected by Theorem 4.2, the **SC-CP** bands adapt to outcome heteroscedasticity within model predictions, while **Marginal** lacks adaptation, **Mondrian** under-adapts due to insufficient bins, and **Kernel** adapts but offers wider intervals for large predictions where observations are sparse. The bands of **CQR** appear adaptive and similar to those of **SC-CP**, however, do not gaurnatee finite-sample prediction-conditional validity. The calibration plots reveal that heteroscedasticity in outcomes is primarily driven by their mean, suggesting that prediction-conditional validity may approximate context-conditional validity. This heuristic is supported by the

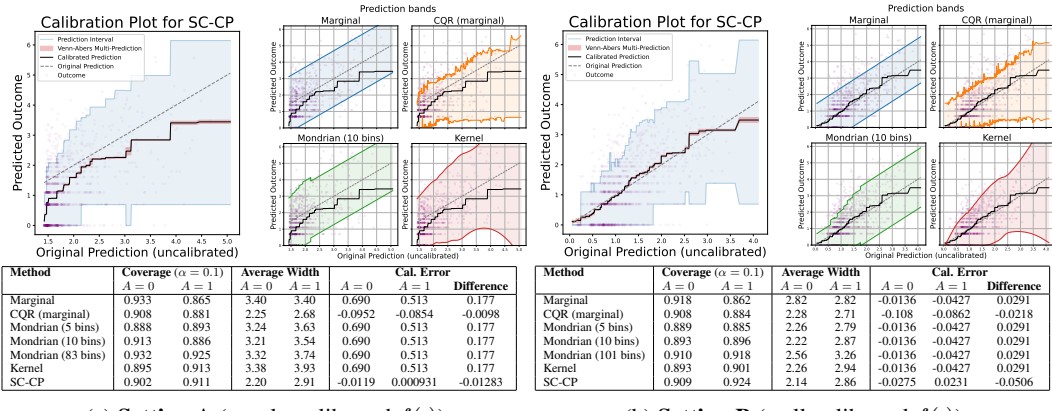

| Method | Coverage ($\alpha = 0.1$) | | Average Width | | Cal. Error | | |
|---|---|---|---|---|---|---|---|
| | $A=0$ | $A=1$ | $A=0$ | $A=1$ | $A=0$ | $A=1$ | Difference |
| Marginal | 0.933 | 0.865 | 3.40 | 3.40 | 0.690 | 0.513 | 0.177 |
| CQR (marginal) | 0.908 | 0.881 | 2.25 | 2.68 | -0.0952 | -0.0854 | -0.0098 |
| Mondrian (5 bins) | 0.888 | 0.893 | 3.24 | 3.63 | 0.690 | 0.513 | 0.177 |
| Mondrian (10 bins) | 0.913 | 0.886 | 3.21 | 3.54 | 0.690 | 0.513 | 0.177 |
| Mondrian (83 bins) | 0.932 | 0.925 | 3.32 | 3.74 | 0.690 | 0.513 | 0.177 |
| Kernel | 0.895 | 0.913 | 3.38 | 3.93 | 0.690 | 0.513 | 0.177 |
| SC-CP | 0.902 | 0.911 | 2.20 | 2.91 | -0.0119 | 0.000931 | -0.01283 |

(a) **Setting A** (poorly-calibrated $f(\cdot)$)

| Method | Coverage ($\alpha = 0.1$) | | Average Width | | Cal. Error | | |
|---|---|---|---|---|---|---|---|
| | $A=0$ | $A=1$ | $A=0$ | $A=1$ | $A=0$ | $A=1$ | Difference |
| Marginal | 0.918 | 0.862 | 2.82 | 2.82 | -0.0136 | -0.0427 | 0.0291 |
| CQR (marginal) | 0.908 | 0.884 | 2.28 | 2.71 | -0.108 | -0.0862 | -0.0218 |
| Mondrian (5 bins) | 0.889 | 0.885 | 2.26 | 2.79 | -0.0136 | -0.0427 | 0.0291 |
| Mondrian (10 bins) | 0.893 | 0.896 | 2.22 | 2.87 | -0.0136 | -0.0427 | 0.0291 |
| Mondrian (101 bins) | 0.910 | 0.918 | 2.56 | 3.26 | -0.0136 | -0.0427 | 0.0291 |
| Kernel | 0.893 | 0.901 | 2.26 | 2.94 | -0.0136 | -0.0427 | 0.0291 |
| SC-CP | 0.909 | 0.924 | 2.14 | 2.86 | -0.0275 | 0.0231 | -0.0506 |

(b) **Setting B** (well-calibrated $f(\cdot)$)

Figure 2: *MEPS-21* **dataset:** Calibration plot for SC-CP, prediction bands for SC-CP and baselines, and empirical coverage, width, and calibration error within sensitive subgroup.

tables in Figure 2, which display empirical coverage, average interval width, and calibration error within the sensitive attribute ($A$) for all methods. In **Setting A**, the base regression model $f(\cdot)$ are poorly calibrated, i.e., $\mathbb{E}[f(X) - Y \mid A]$ is not close to 0, resulting in wider intervals, overconfidence in point predictions, and decreased interpretability for the baselines, as their intervals center around biased point predictions. In contrast, being self-calibrated, **SC-CP** corrects the calibration error in $f$, achieving the smallest interval widths and well-calibrated point predictions in both settings, as guaranteed by Theorem 4.1. In both settings, the quantile regression model of **CQR** appears to have worse calibration than **SC-CP**, which may be due to the median differing from the mean because of the skewness of the outcomes. Additionally, **SC-CP** predictions achieves a smaller difference in calibration error between the two subgroups than **Marginal**, **Mondrian**, and **Kernel**, suggesting they are less discriminatory and more fair [47]. **SC-CP** and **Kernel** achieve the desired coverage level of $1 - \alpha = 0.9$ in each subgroup and setting, whereas **Marginal** exhibits over- or under-coverage in each subgroup. **Mondrian** tends to under-cover with 5 and 10 bins and only attains good coverage when using the same binning number data-adaptively selected by **SC-CP**, highlighting its sensitivity to the pre-specified binning scheme. **CQR** attains good coverage in the $A = 0$ group but undercovers in the $A = 1$ group, which may be explained by **CQR** only guaranteeing marginal coverage in finite samples. Even with **SC-CP** having higher coverage, the intervals of **SC-CP** are narrower than those of **Kernel** and **Mondrian***. This provides experimental evidence that calibration improves conformity scores and translates into greater interval efficiency, as suggested by Theorem 4.3.

## 6 Extensions

Our theoretical techniques can be used to analyze conformal prediction methods that involve the calibration of model predictions followed by the construction of conditionally valid prediction intervals. Our analysis can be adapted to the general case where either the conformity score or the conditioning variable depends on the calibrated model prediction. While we use the absolute residual conformity score in our work, SC-CP can be applied to other conformity scores, such as the normalized absolute residual scoring function [43], allowing for the inclusion of context-specific difficulty estimates in the SC-CP procedure. Although we use Venn-Abers calibration in SC-CP, our analysis also applies to other binning calibration methods, such as Venn calibration [61, 60]. Thus, we can replace the isotonic calibration step in Alg. 1 and 2, for example, with histogram binning [25]. Additionally, a group-valid form of SC-CP can be achieved by applying Alg. 2 separately within subgroups, similar to Multivalid CP [32]. Interesting areas for future work involve integrating point calibration with conformal prediction methods for predictive models beyond regression, such as the isotonic calibration of quantile predictions in conformal quantile regression [48].

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

## A  Code

The methods implemented in this paper are not computationally intensive and were run in a Jupyter notebook environment on a MacBook Pro with 16GB RAM and an M1 chip. A Python implementation of SC-CP is provided in the package `SelfCalibratingConformal`, available via `pip`. Code implementing SC-CP and reproducing our experiments is available in the GitHub repository `SelfCalibratingConformal`, which can be accessed at the following link: `https://github.com/Larsvanderlaan/SelfCalibratingConformal`.

## B  Supplementary real data experiments

### B.1  Additional results

In this section, we present the experimental results for the *concrete*, *STAR*, *bike*, *community*, and *bio* datasets used in [48] and publicly available in the Python package `cqr`, associated with [48] and [47]. For the *STAR* dataset, the sensitive attribute $A$ was set to "gender." For the *Bike* dataset, the sensitive attribute $A$ was set to "workingday," and for *Community*, it was set to "race_binary." For the remaining datasets, the sensitive attribute $A$ was set to a dichotomization of the final column in the feature matrix, as above or below its median value.

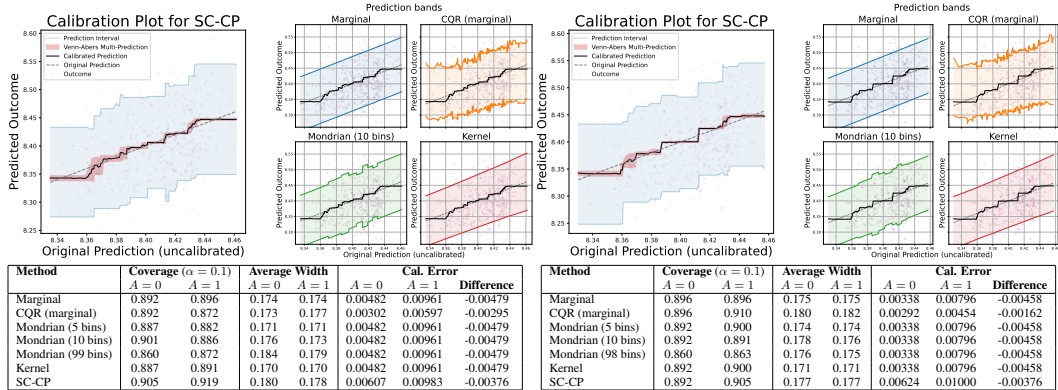

| Method | Coverage ($\alpha=0.1$) | | Average Width | | Cal. Error | | |
|---|---|---|---|---|---|---|---|
| | $A=0$ | $A=1$ | $A=0$ | $A=1$ | $A=0$ | $A=1$ | Difference |
| Marginal | 0.892 | 0.896 | 0.174 | 0.174 | 0.00482 | 0.00961 | -0.00479 |
| CQR (marginal) | 0.892 | 0.872 | 0.173 | 0.177 | 0.00302 | 0.00597 | -0.00295 |
| Mondrian (5 bins) | 0.887 | 0.882 | 0.171 | 0.171 | 0.00482 | 0.00961 | -0.00479 |
| Mondrian (10 bins) | 0.901 | 0.886 | 0.176 | 0.173 | 0.00482 | 0.00961 | -0.00479 |
| Mondrian (99 bins) | 0.860 | 0.872 | 0.184 | 0.179 | 0.00482 | 0.00961 | -0.00479 |
| Kernel | 0.887 | 0.891 | 0.170 | 0.170 | 0.00482 | 0.00961 | -0.00479 |
| SC-CP | 0.905 | 0.919 | 0.180 | 0.178 | 0.00607 | 0.00983 | -0.00376 |

| Method | Coverage ($\alpha=0.1$) | | Average Width | | Cal. Error | | |
|---|---|---|---|---|---|---|---|
| | $A=0$ | $A=1$ | $A=0$ | $A=1$ | $A=0$ | $A=1$ | Difference |
| Marginal | 0.896 | 0.896 | 0.175 | 0.175 | 0.00338 | 0.00796 | -0.00458 |
| CQR (marginal) | 0.896 | 0.910 | 0.180 | 0.182 | 0.00292 | 0.00454 | -0.00162 |
| Mondrian (5 bins) | 0.892 | 0.900 | 0.174 | 0.174 | 0.00338 | 0.00796 | -0.00458 |
| Mondrian (10 bins) | 0.892 | 0.891 | 0.178 | 0.176 | 0.00338 | 0.00796 | -0.00458 |
| Mondrian (98 bins) | 0.860 | 0.863 | 0.176 | 0.175 | 0.00338 | 0.00796 | -0.00458 |
| Kernel | 0.892 | 0.900 | 0.171 | 0.171 | 0.00338 | 0.00796 | -0.00458 |
| SC-CP | 0.892 | 0.905 | 0.177 | 0.177 | 0.00624 | 0.01000 | -0.00376 |

(a) **Setting A** (poorly-calibrated $f(\cdot)$)   (b) **Setting B** (well-calibrated $f(\cdot)$)

Figure 3: ***STAR* dataset:** Calibration plot for SC-CP, prediction bands for SC-CP and baselines, and empirical coverage, width, and calibration error within sensitive subgroup.

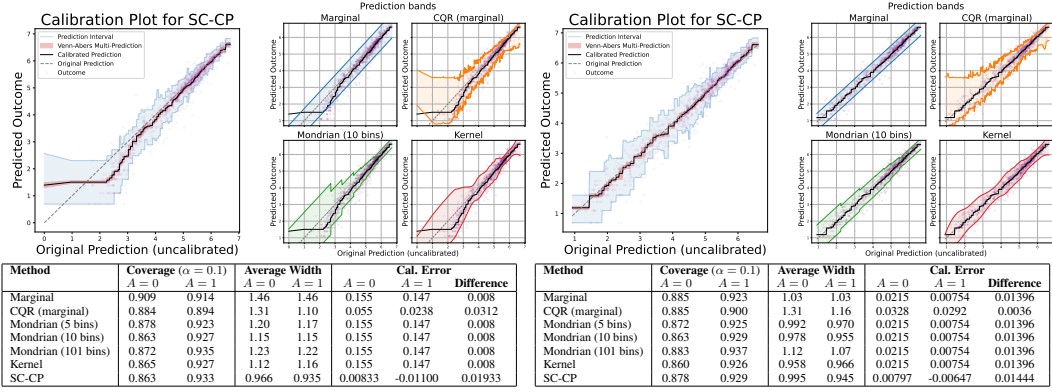

| Method | Coverage ($\alpha=0.1$) | | Average Width | | Cal. Error | | |
|---|---|---|---|---|---|---|---|
| | $A=0$ | $A=1$ | $A=0$ | $A=1$ | $A=0$ | $A=1$ | Difference |
| Marginal | 0.909 | 0.914 | 1.46 | 1.46 | 0.155 | 0.147 | 0.008 |
| CQR (marginal) | 0.884 | 0.894 | 1.31 | 1.10 | 0.055 | 0.0238 | 0.0312 |
| Mondrian (5 bins) | 0.878 | 0.923 | 1.20 | 1.17 | 0.155 | 0.147 | 0.008 |
| Mondrian (10 bins) | 0.863 | 0.927 | 1.15 | 1.15 | 0.155 | 0.147 | 0.008 |
| Mondrian (101 bins) | 0.872 | 0.935 | 1.23 | 1.22 | 0.155 | 0.147 | 0.008 |
| Kernel | 0.865 | 0.927 | 1.12 | 1.16 | 0.155 | 0.147 | 0.008 |
| SC-CP | 0.863 | 0.933 | 0.966 | 0.935 | 0.00833 | -0.01100 | 0.01933 |

| Method | Coverage ($\alpha=0.1$) | | Average Width | | Cal. Error | | |
|---|---|---|---|---|---|---|---|
| | $A=0$ | $A=1$ | $A=0$ | $A=1$ | $A=0$ | $A=1$ | Difference |
| Marginal | 0.885 | 0.923 | 1.03 | 1.03 | 0.0215 | 0.00754 | 0.01396 |
| CQR (marginal) | 0.885 | 0.900 | 1.31 | 1.16 | 0.0328 | 0.0292 | 0.0036 |
| Mondrian (5 bins) | 0.872 | 0.925 | 0.992 | 0.970 | 0.0215 | 0.00754 | 0.01396 |
| Mondrian (10 bins) | 0.863 | 0.929 | 0.978 | 0.955 | 0.0215 | 0.00754 | 0.01396 |
| Mondrian (101 bins) | 0.883 | 0.937 | 1.12 | 1.07 | 0.0215 | 0.00754 | 0.01396 |
| Kernel | 0.860 | 0.926 | 0.958 | 0.966 | 0.0215 | 0.00754 | 0.01396 |
| SC-CP | 0.878 | 0.929 | 0.995 | 0.945 | 0.00797 | -0.00647 | 0.01444 |

(a) **Setting A** (poorly-calibrated $f(\cdot)$)   (b) **Setting B** (well-calibrated $f(\cdot)$)

Figure 4: ***Bike* dataset:** Calibration plot for SC-CP, prediction bands for SC-CP and baselines, and empirical coverage, width, and calibration error within sensitive subgroup.

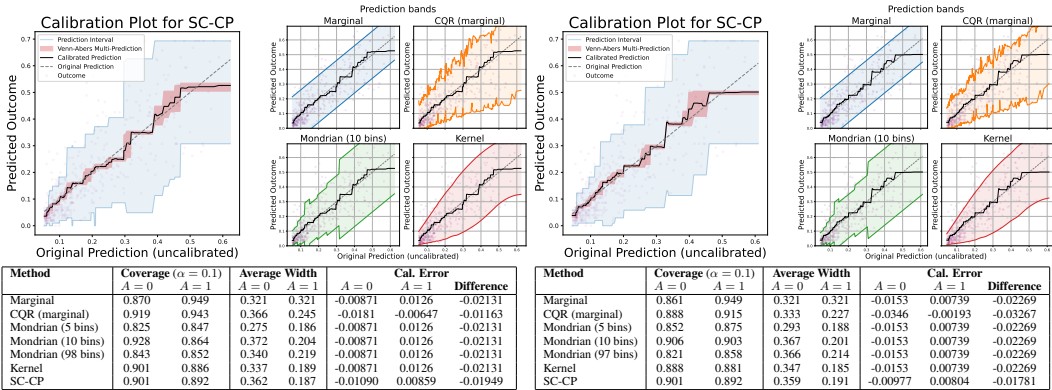

| Method | Coverage ($\alpha = 0.1$) | | Average Width | | Cal. Error | | |
|---|---|---|---|---|---|---|---|
| | $A = 0$ | $A = 1$ | $A = 0$ | $A = 1$ | $A = 0$ | $A = 1$ | **Difference** |
| Marginal | 0.870 | 0.949 | 0.321 | 0.321 | -0.00871 | 0.0126 | -0.02131 |
| CQR (marginal) | 0.919 | 0.943 | 0.366 | 0.245 | -0.0181 | -0.00647 | -0.01163 |
| Mondrian (5 bins) | 0.825 | 0.847 | 0.275 | 0.186 | -0.00871 | 0.0126 | -0.02131 |
| Mondrian (10 bins) | 0.928 | 0.864 | 0.372 | 0.204 | -0.00871 | 0.0126 | -0.02131 |
| Mondrian (98 bins) | 0.843 | 0.852 | 0.340 | 0.219 | -0.00871 | 0.0126 | -0.02131 |
| Kernel | 0.901 | 0.886 | 0.337 | 0.189 | -0.00871 | 0.0126 | -0.02131 |
| SC-CP | 0.901 | 0.892 | 0.362 | 0.187 | -0.01090 | 0.00859 | -0.01949 |

| Method | Coverage ($\alpha = 0.1$) | | Average Width | | Cal. Error | | |
|---|---|---|---|---|---|---|---|
| | $A = 0$ | $A = 1$ | $A = 0$ | $A = 1$ | $A = 0$ | $A = 1$ | **Difference** |
| Marginal | 0.861 | 0.949 | 0.321 | 0.321 | -0.0153 | 0.00739 | -0.02269 |
| CQR (marginal) | 0.888 | 0.915 | 0.333 | 0.227 | -0.0346 | -0.00193 | -0.03267 |
| Mondrian (5 bins) | 0.852 | 0.875 | 0.293 | 0.188 | -0.0153 | 0.00739 | -0.02269 |
| Mondrian (10 bins) | 0.906 | 0.903 | 0.367 | 0.201 | -0.0153 | 0.00739 | -0.02269 |
| Mondrian (97 bins) | 0.821 | 0.858 | 0.366 | 0.214 | -0.0153 | 0.00739 | -0.02269 |
| Kernel | 0.888 | 0.881 | 0.347 | 0.185 | -0.0153 | 0.00739 | -0.02269 |
| SC-CP | 0.901 | 0.892 | 0.359 | 0.191 | -0.00977 | 0.00804 | -0.01781 |

(a) **Setting A** (poorly-calibrated $f(\cdot)$)      (b) **Setting B** (well-calibrated $f(\cdot)$)

Figure 5: *Community* **dataset:** Calibration plot for SC-CP, prediction bands for SC-CP and baselines, and empirical coverage, width, and calibration error within sensitive subgroup.

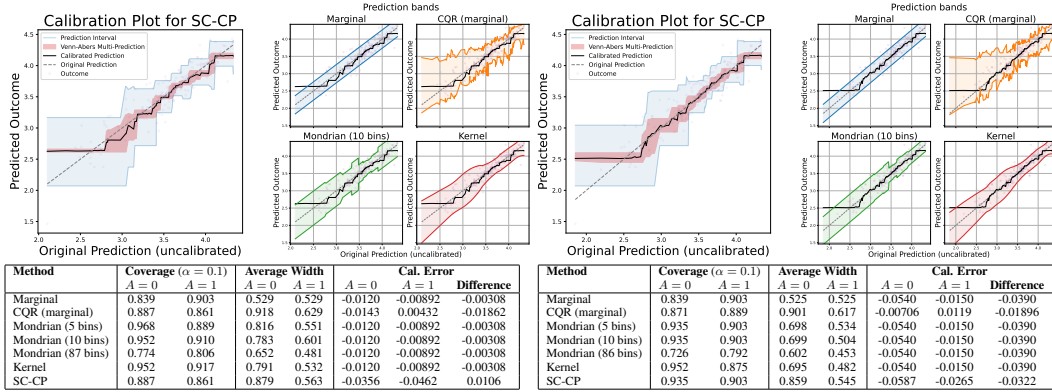

| Method | Coverage ($\alpha = 0.1$) | | Average Width | | Cal. Error | | |
|---|---|---|---|---|---|---|---|
| | $A = 0$ | $A = 1$ | $A = 0$ | $A = 1$ | $A = 0$ | $A = 1$ | **Difference** |
| Marginal | 0.839 | 0.903 | 0.529 | 0.529 | -0.0120 | -0.00892 | -0.00308 |
| CQR (marginal) | 0.887 | 0.861 | 0.918 | 0.629 | -0.0143 | 0.00432 | -0.01862 |
| Mondrian (5 bins) | 0.968 | 0.889 | 0.816 | 0.551 | -0.0120 | -0.00892 | -0.00308 |
| Mondrian (10 bins) | 0.952 | 0.910 | 0.783 | 0.601 | -0.0120 | -0.00892 | -0.00308 |
| Mondrian (87 bins) | 0.774 | 0.806 | 0.652 | 0.481 | -0.0120 | -0.00892 | -0.00308 |
| Kernel | 0.952 | 0.917 | 0.791 | 0.532 | -0.0120 | -0.00892 | -0.00308 |
| SC-CP | 0.887 | 0.861 | 0.879 | 0.563 | -0.0356 | -0.0462 | 0.0106 |

| Method | Coverage ($\alpha = 0.1$) | | Average Width | | Cal. Error | | |
|---|---|---|---|---|---|---|---|
| | $A = 0$ | $A = 1$ | $A = 0$ | $A = 1$ | $A = 0$ | $A = 1$ | **Difference** |
| Marginal | 0.839 | 0.903 | 0.525 | 0.525 | -0.0540 | -0.0150 | -0.0390 |
| CQR (marginal) | 0.871 | 0.889 | 0.901 | 0.617 | -0.00706 | 0.0119 | -0.01896 |
| Mondrian (5 bins) | 0.935 | 0.903 | 0.698 | 0.534 | -0.0540 | -0.0150 | -0.0390 |
| Mondrian (10 bins) | 0.935 | 0.903 | 0.699 | 0.504 | -0.0540 | -0.0150 | -0.0390 |
| Mondrian (86 bins) | 0.726 | 0.792 | 0.602 | 0.453 | -0.0540 | -0.0150 | -0.0390 |
| Kernel | 0.952 | 0.875 | 0.695 | 0.482 | -0.0540 | -0.0150 | -0.0390 |
| SC-CP | 0.935 | 0.903 | 0.545 | 0.545 | -0.0587 | -0.0265 | -0.0322 |

(a) **Setting A** (poorly-calibrated $f(\cdot)$)      (b) **Setting B** (well-calibrated $f(\cdot)$)

Figure 6: *Concrete* **dataset:** Calibration plot for SC-CP, prediction bands for SC-CP and baselines, and empirical coverage, width, and calibration error within sensitive subgroup.

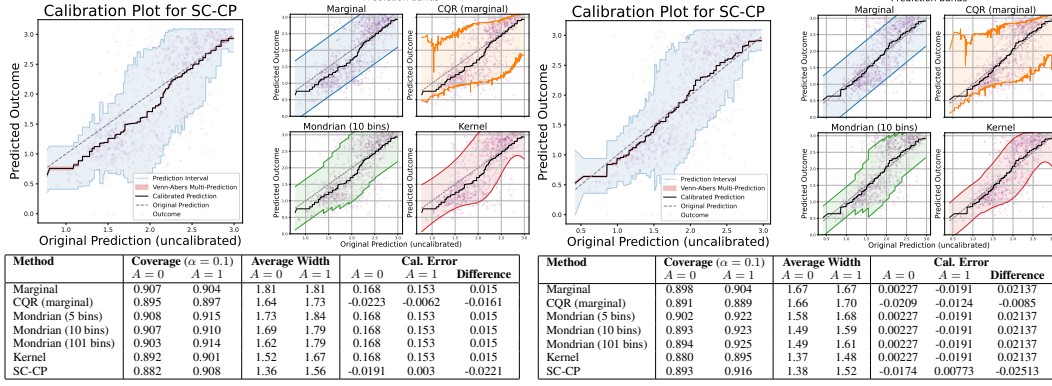

| Method | Coverage ($\alpha = 0.1$) | | Average Width | | Cal. Error | | |
|---|---|---|---|---|---|---|---|
| | $A = 0$ | $A = 1$ | $A = 0$ | $A = 1$ | $A = 0$ | $A = 1$ | **Difference** |
| Marginal | 0.907 | 0.904 | 1.81 | 1.81 | 0.168 | 0.153 | 0.015 |
| CQR (marginal) | 0.895 | 0.897 | 1.64 | 1.73 | -0.0223 | -0.0062 | -0.0161 |
| Mondrian (5 bins) | 0.908 | 0.915 | 1.73 | 1.84 | 0.168 | 0.153 | 0.015 |
| Mondrian (10 bins) | 0.907 | 0.910 | 1.69 | 1.79 | 0.168 | 0.153 | 0.015 |
| Mondrian (101 bins) | 0.903 | 0.914 | 1.62 | 1.79 | 0.168 | 0.153 | 0.015 |
| Kernel | 0.892 | 0.901 | 1.52 | 1.67 | 0.168 | 0.153 | 0.015 |
| SC-CP | 0.882 | 0.908 | 1.36 | 1.56 | -0.0191 | 0.003 | -0.0221 |

| Method | Coverage ($\alpha = 0.1$) | | Average Width | | Cal. Error | | |
|---|---|---|---|---|---|---|---|
| | $A = 0$ | $A = 1$ | $A = 0$ | $A = 1$ | $A = 0$ | $A = 1$ | **Difference** |
| Marginal | 0.898 | 0.904 | 1.67 | 1.67 | 0.00227 | -0.0191 | 0.02137 |
| CQR (marginal) | 0.891 | 0.889 | 1.66 | 1.70 | -0.0209 | -0.0124 | -0.0085 |
| Mondrian (5 bins) | 0.902 | 0.922 | 1.58 | 1.68 | 0.00227 | -0.0191 | 0.02137 |
| Mondrian (10 bins) | 0.893 | 0.923 | 1.49 | 1.59 | 0.00227 | -0.0191 | 0.02137 |
| Mondrian (101 bins) | 0.894 | 0.925 | 1.49 | 1.61 | 0.00227 | -0.0191 | 0.02137 |
| Kernel | 0.880 | 0.895 | 1.37 | 1.48 | 0.00227 | -0.0191 | 0.02137 |
| SC-CP | 0.893 | 0.916 | 1.38 | 1.52 | -0.0174 | 0.00773 | -0.02513 |

(a) **Setting A** (poorly-calibrated $f(\cdot)$)      (b) **Setting B** (well-calibrated $f(\cdot)$)

Figure 7: *Bio* **dataset:** Calibration plot for SC-CP, prediction bands for SC-CP and baselines, and empirical coverage, width, and calibration error within sensitive subgroup.

## C   Supplementary synthetic data experiments

### C.1   Experimental setup

In this appendix, we perform additional synthetic experiments to evaluate the prediction-conditional validity of our method and how it translates to approximate context-conditional validity in certain cases.

**Synthetic datasets.** We construct synthetic training, calibration, and test datasets $\mathcal{D}_{train}, \mathcal{D}_{cal}, \mathcal{D}_{test}$ of sizes $n_{train}, n_{cal}, n_{test}$, which are respectively used to train $f$, apply CP, and evaluate performance. For parameters $d \in \mathbb{N}, \kappa > 0, a \geq 0, b \geq 0$, each dataset consists of *iid* observations of $(X, Y)$ drawn as follows. The covariate vector $X := (X_1, \ldots, X_d) \in [0,1]^d$ is coordinate-wise independently drawn from a Beta$(1, \kappa)$ distribution with shape parameter $\kappa$. Then, conditionally on $X = x$, the outcome $Y$ is drawn normally distributed with conditional mean $\mu(x) := d^{-1/2} \sum_{j=1}^{d} \{x_j + \sin(4x_j)\}$ and conditional variance $\sigma^2(x) := \{0.035 - a \log(0.5 + 0.5x_1)/8 + b\left(|\mu_0(x)|^6/20 - 0.02\right)/2\}^2$. Here, $a$ and $b$ control the heteroscedasticity and mean-variance relationship for the outcomes. For $\mathcal{D}_{cal}$ and $\mathcal{D}_{test}$, we set $\kappa_{cal} = \kappa_{test} = 1$ and, for $\mathcal{D}_{train}$, we vary $\kappa_{train}$ to introduce distribution shift and, thereby, calibration error in $f$. The parameters $d$, $a$, and $b$ are fixed across the datasets.

**Baselines.** To mitigate overfitting, we implement SC-CP so that each function in $\Theta_{iso}$ has at least 20 observations averaged within each constant segment (implemented using the minimum leaf node size of argument xgboost). When appropriate, we will consider the following baseline CP algorithms for comparison. Unless stated otherwise, for all baselines, we use the scoring function $S(x, y) := |y - f(x)|$. The first baseline, uncond-CP, is split-CP [36], which provides only unconditional coverage guarantees. The second baseline, cond-CP, is adapted from [21] and provides conditional coverage over distribution shifts within a specified reproducing kernel Hilbert space. Following Section 5.1 of [21], we use the Gaussian kernel $K(X_i, X_j) := \exp\left(-4\|X_i - X_j\|_2^2\right)$ with euclidean norm $\|\cdot\|_2$ and select the regularization parameter $\lambda$ using 5-fold cross-validation. The third baseline, Mondrian CP, applies the Mondrian CP method [59, 8] to categories formed by dividing $f$'s predictions into 20 equal-frequency bins based on $\mathcal{D}_{cal}$. As an optimal benchmark, we consider the oracle satisfying (2).

## C.2  Experiment 1: Calibration and efficiency

In this experiment, we illustrate how calibration of the predictor $f$ can improve the efficiency (i.e., width) of the resulting prediction intervals. We consider the data-generating process of the previous section, with $n_{train} = n_{cal} = n_{test} = 1000$, $d = 5$, $a = 0$, and $b = 0.6$. The predictor $f$ is trained on $\mathcal{D}_{train}$ using the *ranger* [63] implementation of random forests with default settings. To control the calibration error in $f$, we vary the distribution shift parameter $\kappa_{train}$ for $\mathcal{D}_{train}$ over $\{1, 1.5, 2, 2.5, 3\}$.

**Results.** Figure 8a compares the average interval width across $\mathcal{D}_{test}$ for SC-CP and baselines as the $\ell^2$-calibration error in $f$ increases. Here, we estimate the calibration error using the approach of [64]. As calibration error increases, the average interval width for SC-SP appears smaller than those of uncond-CP, Mondrian-CP, and cond-CP, especially in comparison to uncond-CP. The observed efficiency gains in SC-CP are consistent with Theorem 4.3 and provides empirical evidence that self-calibrated conformity scores translate to tighter prediction intervals, given sufficient data. To test this hypothesis under controlled conditions, we compare the widths of prediction intervals obtained using vanilla (unconditional) CP for two scoring functions: $S : (x, y) \mapsto |y - f(x)|$ and the Venn-Abers (worst-case) scoring function $S_{\text{cal}} : (x, y) \mapsto \max_{y \in \mathcal{Y}} |y - f_n^{(x,y)}(x)|$. For miscoverage levels $\alpha \in \{0.05, 0.1, 0.2\}$, the left panel of Figure 8b illustrates the relative efficiency gain achieved by using $S_{\text{cal}}$, which we define as the ratio of the average interval widths for $S_{\text{cal}}$ relative to $S$. The widths and calibration errors in Figure 8b are averaged across 100 data replicates.

**Role of calibration set size.** With too small calibration sets, the isotonic calibration step in Alg. 2 can lead to overfitting. In such cases, the self-calibrated conformity scores could be larger than their uncalibrated counterparts, potentially resulting in less efficient prediction intervals. In our experiments, overfitting is mitigated by constraining the minimum size of the leaf node in the isotonic regression tree to 20. For $\alpha \in \{0.05, 0.1, 0.2\}$, the right panel of Figure 8b displays the relationship between $n_{cal}$ and the relative efficiency gain achieved by using $S_{\text{cal}}$, holding calibration error fixed ($\kappa_{train} = 3$). We find that calibration leads to a noticeable reduction in interval width as soon as $n_{cal} \geq 50$.

## C.3  Experiment 2: Coverage and Adaptivity

In this experiment, we illustrate how self-calibration of $\widehat{C}_{n+1}(X_{n+1})$ can, in some cases, translate to stronger conditional coverage guarantees. Here we take $n_{train} = n_{cal} = 1000$ and $n_{test} = 2500$,

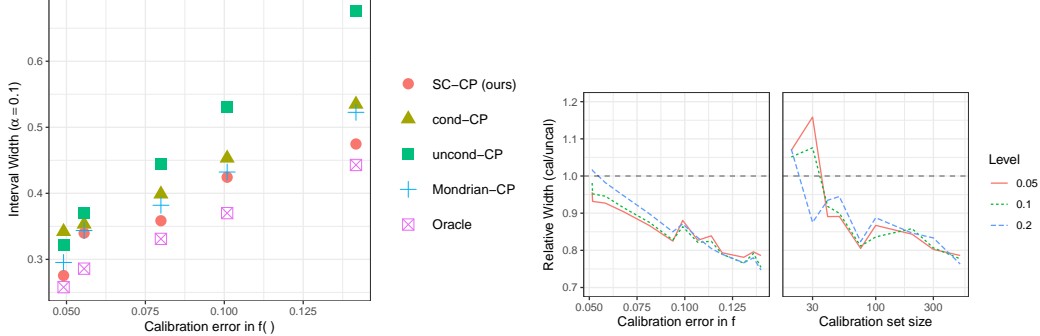

(a) Avg. interval width with varying calibration error.   (b) Relative efficiency change from calibration.

Figure 8: Figure 8a shows the average interval widths for varying $\ell^2$-calibration errors in $f$. Figure 8b shows the relative change in average interval width using Venn-Abers calibrated versus uncalibrated predictions, with varying $\ell^2$-calibration error in $f$ (left), and calibration set size (right). Below the horizontal lines signifies efficiency gains for SC-CP.

and no distribution shift ($\kappa_{train} = 1$) in $\mathcal{D}_{train}$. We consider three setups: **Setup A** ($d = 5, a = 0, b = 0.6$) and **Setup B** ($d = 5, a = 0, b = 0.6$) which have a strong mean-variance relationship in the outcome process; and **Setup C** ($d = 5, a = 0.6, b = 0$) which has no such relationship. We obtain the predictor $f$ from $\mathcal{D}_{train}$ using a generalized additive model [27], so that it is well calibrated and accurately estimates $\mu$. To assess the adaptivity of SC-CP to heteroscedasticity, we report the coverage and the average interval width within subgroups of $\mathcal{D}_{test}$ defined by quintiles of the conditional variances $\{\sigma^2(X_i) : (X_i, Y_i) \in \mathcal{D}_{test}\}$.

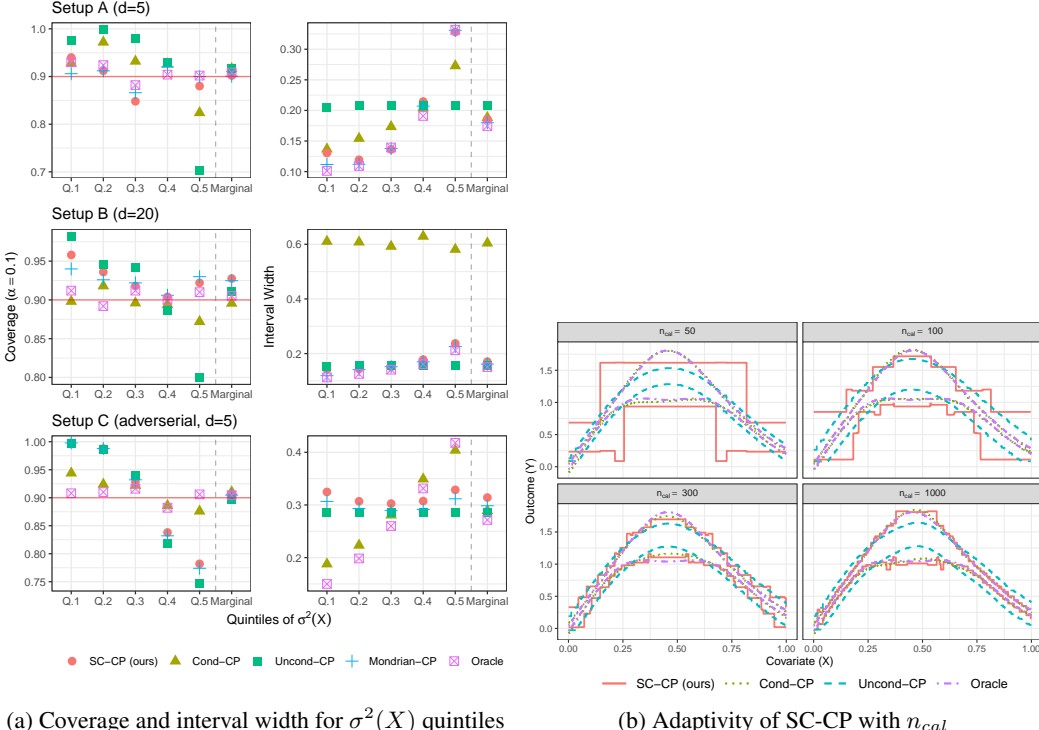

(a) Coverage and interval width for $\sigma^2(X)$ quintiles   (b) Adaptivity of SC-CP with $n_{cal}$

Figure 9: Figure 9a displays the coverage and average interval width of SC-CP, marginally and within quintiles of the conditional outcome variance for setups A, B, and C. For a $d = 1$ example, Figure 9b shows the adaptivity of SC-CP prediction bands ($\alpha = 0.1$) across various calibration set sizes.

**Results.** The top two panels in Figure 9a displays the coverage and average interval width results for **Setup A** and **Setup B**. In both setups, SC-CP, cond-CP, and Mondrian-CP exhibit satisfactory coverage both marginally and within the quintile subgroups. In contrast, while uncond-CP attains the nominal level of marginal coverage, it exhibits noticeable overcoverage within the first three quintiles and significant undercoverage within the fifth quintile. The satisfactory performance of SC-CP with respect to heteroscedasticity in **Setups A and B** can be attributed to the strong mean-variance relationship in the outcome process. Regarding efficiency, the average interval widths of SC-CP are competitive with those of prediction-binned Mondrian-CP and the oracle intervals. The interval widths for cond-CP are wider than those for SC-CP and Mondrian-CP, especially for **Setup B**. This difference can be explained by cond-CP aiming for conditional validity in a 5D and 20D space, whereas SC-CP and Mondrian-CP target the 1D output space.

**Limitation.** If there is no mean-variance relationship in the outcomes, SC-CP is generally not expected to adapt to heteroscedasticity. The third (bottom) panel of Figure 9a displays SC-CP's performance in **Setup C**, where there is no such relationship. In this scenario, it is evident that the conditional coverage of both uncond-CP and SC-CP are poor, while cond-CP maintains adequate coverage.

**Adaptivity and calibration set size.** SC-CP can be derived by applying CP within subgroups defined by a data-dependent binning of the output space $f(\mathcal{X})$, learned via Venn-Abers calibration, where the number of bins can grow with $n_{cal}$. In particular, if $t \mapsto E[Y \mid f(X) = t]$ is asymptotically monotone, which is plausible when $f$ consistently estimates the outcome regression, then the number of bins selected by isotonic calibration will generally increase with $n_{cal}$, and the width of these bins will tend to zero. As a consequence, in such cases, the self-calibration result in Theorem 4.2 translates to conditional guarantees over finer partitions of $f(\mathcal{X})$ as $n_{cal}$ increases. For $d = 1$ and a strong mean-variance relationship ($a = 0, b = 0.6$), Figure 9b demonstrates how the adaptivity of SC-CP bands improves as $n_{cal}$ increases. In this case, for $n_{cal}$ sufficiently large, we find that the SC-CP bands closely match those of cond-CP and the oracle.

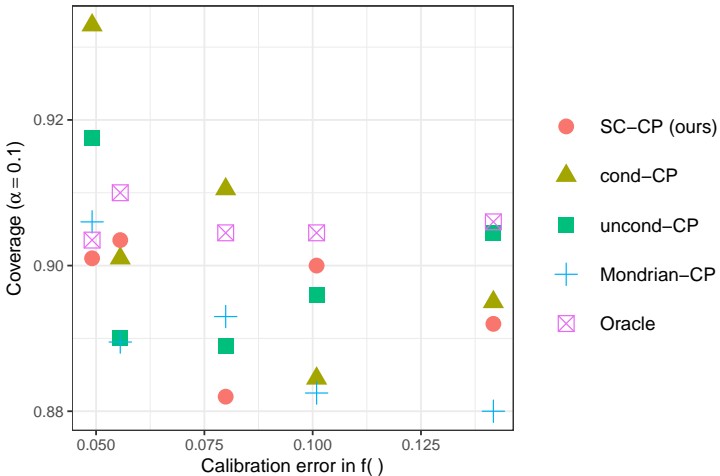

(a) Avg. interval width with varying calibration error.

Figure 10: Figure 10a shows the marginal coverage corresponding to the interval widths of Figure 8a for varying $\ell^2$-calibration errors in $f$.

# D   Proofs

*Proof of Theorem 4.1.* It can be verified that an *isotonic calibrator class* $\Theta_{\text{iso}}$ satisfies the following properties: (a) It consists of univariate regression trees (i.e., piecewise constant functions) that are monotonically nondecreasing; (b) For all elements $\theta \in \Theta_{\text{iso}}$ and transformations $g : \mathbb{R} \to \mathbb{R}$, it holds that $g \circ \theta \in \Theta_{\text{iso}}$. Property (a) holds by definition. Property (b) is satisfied since constraining the maximum number of constant segments by $k(n)$ does not constrain the possible values that the

function may take in a given constant region. We note that the result of this theorem holds in general for any function class $\Theta_{\text{iso}}$ that satisfies (a) and (b).

Recall from Alg. 1 that $f_n^{(X_{n+1},Y_{n+1})} = \theta_n^{(X_{n+1},Y_{n+1})} \circ f$, where $\theta_n^{(X_{n+1},Y_{n+1})} \in \Theta_{iso}$ is an isotonic calibrator satisfying $f_n^{(X_{n+1},Y_{n+1})} = \theta_n^{(X_{n+1},Y_{n+1})} \circ f$. By definition, recall that the isotonic calibrator class $\Theta_{iso}$ satisfies the invariance property that, for all $g : \mathbb{R} \to \mathbb{R}$, the inclusion $\theta \in \Theta_{iso}$ implies $g \circ \theta \in \Theta_{iso}$. Hence, for all $g : \mathbb{R} \to \mathbb{R}$ and $\varepsilon > 0$, it also holds that $(1 + \varepsilon g) \circ \theta_n^{(X_{n+1},Y_{n+1})}$ lies in $\Theta_{iso}$. Now, we use that $(1 + \varepsilon g) \circ \theta_n^{(X_{n+1},Y_{n+1})} \circ f = (1 + \varepsilon g) \circ f_n^{(X_{n+1},Y_{n+1})}$ and that $f_n^{(X_{n+1},Y_{n+1})}$ is an empirical risk minimizer over the class $\{g \circ f : g \in \Theta_{iso}\}$. Using these two observations, the first order derivative equations characterizing the empirical risk minimizer $f_n^{(X_{n+1},Y_{n+1})}$ imply, for all $g : \mathbb{R} \to \mathbb{R}$, that

$$\frac{1}{n+1} \sum_{i=1}^{n+1} (g \circ f_n^{(X_{n+1},Y_{n+1})})(X_i) \left\{ Y_i - f_n^{(X_{n+1},Y_{n+1})}(X_i) \right\} = \frac{d}{d\varepsilon} \left[ \frac{1}{n+1} \sum_{i=1}^{n+1} \left\{ Y_i - (1 + \varepsilon g) \circ f_n^{(X_{n+1},Y_{n+1})} \right\}^2 \right] \Bigg|_{\varepsilon=0}$$

$$= 0. \tag{5}$$

Taking expectations of both sides of the above display, we conclude

$$\frac{1}{n+1} \sum_{i=1}^{n+1} \mathbb{E}\left[ (g \circ f_n^{(X_{n+1},Y_{n+1})})(X_i) \left\{ Y_i - f_n^{(X_{n+1},Y_{n+1})}(X_i) \right\} \right] = 0. \tag{6}$$

We now use the fact that $\{(X_i, Y_i, f_n^{(X_{n+1},Y_{n+1})}(X_i)) : i \in [n+1]\}$ are exchangeable, since $\{(X_i, Y_i)\}_{i=1}^{n+1}$ are exchangeable by C1 and the function $f_n^{(X_{n+1},Y_{n+1})}$ is invariant under permutations of $\{(X_i, Y_i)\}_{i=1}^{n+1}$. Consequently, Equation (6) remains true if we replace each $(X_i, Y_i, f_n^{(X_{n+1},Y_{n+1})}(X_i))$ with $i \in [n]$ by $(X_{n+1}, Y_{n+1}, f_n^{(X_{n+1},Y_{n+1})}(X_{n+1}))$. That is,

$$\mathbb{E}\left[ (g \circ f_n^{(X_{n+1},Y_{n+1})})(X_{n+1}) \left\{ Y_{n+1} - f_n^{(X_{n+1},Y_{n+1})}(X_{n+1}) \right\} \right]$$

$$= \frac{1}{n+1} \sum_{i=1}^{n+1} \mathbb{E}\left[ (g \circ f_n^{(X_{n+1},Y_{n+1})})(X_{n+1}) \left\{ Y_{n+1} - f_n^{(X_{n+1},Y_{n+1})}(X_{n+1}) \right\} \right]$$

$$= \frac{1}{n+1} \sum_{i=1}^{n+1} \mathbb{E}\left[ (g \circ f_n^{(X_{n+1},Y_{n+1})})(X_i) \left\{ Y_i - f_n^{(X_{n+1},Y_{n+1})}(X_i) \right\} \right]$$

$$= 0.$$

By the law of iterated conditional expectations, the preceding display further implies

$$\mathbb{E}\left[ (g \circ f_n^{(X_{n+1},Y_{n+1})})(X_{n+1}) \left\{ \mathbb{E}[Y_{n+1} \mid f_n^{(X_{n+1},Y_{n+1})}(X_{n+1})] - f_n^{(X_{n+1},Y_{n+1})}(X_{n+1}) \right\} \right] = 0.$$

Taking $g : \mathbb{R} \to \mathbb{R}$ to be defined by $(g \circ f_n^{(X_{n+1},Y_{n+1})})(X_{n+1}) := \mathbb{E}[Y_{n+1} \mid f_n^{(X_{n+1},Y_{n+1})}(X_{n+1})] - f_n^{(X_{n+1},Y_{n+1})}(X_{n+1})$, we find

$$\mathbb{E}\left[ \left\{ \mathbb{E}[Y_{n+1} \mid f_n^{(X_{n+1},Y_{n+1})}(X_{n+1})] - f_n^{(X_{n+1},Y_{n+1})}(X_{n+1}) \right\}^2 \right] = 0.$$

The above equality implies $\mathbb{E}[Y_{n+1} \mid f_n^{(X_{n+1},Y_{n+1})}(X_{n+1})] = f_n^{(X_{n+1},Y_{n+1})}(X_{n+1})$ almost surely, as desired.

$\square$

*Proof of Theorem 4.2.* Recall, for a quantile level $\alpha \in (0, 1)$, the "pinball" quantile loss function $\ell_\alpha$ is given by

$$\ell_\alpha(f(x), s) := \begin{cases} \alpha(s - f(x)) & \text{if } s \geq f(x), \\ (1 - \alpha)(f(x) - s) & \text{if } s < f(x). \end{cases}$$

As established in [21], each subgradient of $\ell_\alpha(\cdot, x)$ at $f$ in the direction $g$, for some $\beta \in [\alpha - 1, \alpha]$, given by:

$$\partial_{\varepsilon[\beta]} \left\{ \ell_\alpha(f(x) + \varepsilon g(x), s) \right\} \big|_{\varepsilon=0} := 1(f(x) \neq s)g(x)\{\alpha - 1(f < s)\} + 1(f(x) = s)\beta g(x).$$

For $(x, y) \in \mathcal{X} \times \mathcal{Y}$, define the empirical risk minimizer:

$$\rho_n^{(x,y)} \in \operatorname*{argmin}_{\theta \circ f_n^{(x,y)}; \theta: \mathbb{R} \to \mathbb{R}} \sum_{i=1}^{n} \ell_\alpha \left( \theta \circ f_n^{(x,y)}(X_i), S_i^{(x,y)} \right) + \ell_\alpha \left( \theta \circ f_n^{(x,y)}(x), S_{n+1}^{(x,y)} \right).$$

Then, since the isotonic calibrated predictor $f_n^{(x,y)}$ is piece-wise constant and the above optimization problem is unconstrained in the map $\theta : \mathbb{R} \to \mathbb{R}$, it holds that the evaluation $\rho_n^{(x,y)}(x)$ lies in the solution set:

$$\operatorname*{argmin}_{q \in \mathbb{R}} \sum_{i=1}^{n} K_i(f_n^{(x,y)}, x) \cdot \ell_\alpha(q, S_i^{(x,y)}) + \ell_\alpha(q, S_{n+1}^{(x,y)}),$$

where $K_i(f_n^{(x,y)}, x) = \mathbf{1}\left\{ f_n^{(x,y)}(X_i) = f_n^{(x,y)}(x) \right\}$. Consequently, we see that the empirical quantile $\rho_n^{(x,y)}(x)$ defined in Alg. 2 coincides with the evaluation of the empirical risk minimizer $\rho_n^{(x,y)}(\cdot)$ at $x$, as suggested by our notation.

We will now theoretically analyze Alg. 2 by studying the empirical risk minimizer $x' \mapsto \rho_n^{(X_{n+1}, Y_{n+1})}(x')$. To do so, we modify the arguments used to establish Theorem 2 of [21]. As in [21], we begin by examining the first-order equations of the convex optimization problem defining $x' \mapsto \rho_n^{(X_{n+1}, Y_{n+1})}(x')$.

**Studying first-order equations of convex problem:** Given any transformation $\theta : \mathbb{R} \to \mathbb{R}$, each subgradient of the map

$$\varepsilon \mapsto \ell_\alpha \left( \rho_n^{(X_{n+1}, Y_{n+1})}(X_i) + \varepsilon \theta \circ f_n^{(X_{n+1}, Y_{n+1})}(X_i), S_i^{(X_{n+1}, Y_{n+1})} \right)$$

is, for some $\beta \in [\alpha - 1, \alpha]^{n+1}$, of the following form:

$$\partial_{\varepsilon[\beta]} \left\{ \ell_\alpha \left( \rho_n^{(X_{n+1}, Y_{n+1})}(X_i) + \varepsilon \theta \circ f_n^{(X_{n+1}, Y_{n+1})}(X_i), S_i^{(X_{n+1}, Y_{n+1})} \right) \right\} \Big|_{\varepsilon=0}$$

$$:= \begin{cases} \theta \circ f_n^{(X_{n+1}, Y_{n+1})}(X_i)\{\alpha - 1(\rho_n^{(X_{n+1}, Y_{n+1})}(X_i) < S_i^{(X_{n+1}, Y_{n+1})})\} & \text{if } S_i^{(X_{n+1}, Y_{n+1})} \neq \rho_n^{(X_{n+1}, Y_{n+1})}(X_i), \\ \beta_i(\theta \circ f_n^{(X_{n+1}, Y_{n+1})})(X_i) & \text{if } S_i^{(X_{n+1}, Y_{n+1})} = \rho_n^{(X_{n+1}, Y_{n+1})}(X_i). \end{cases}$$

Now, since $\rho_n^{(X_{n+1}, Y_{n+1})}$ is an empirical risk minimizer of the quantile loss over the class $\{\theta \circ f_n^{(X_{n+1}, Y_{n+1})}; \theta : \mathbb{R} \to \mathbb{R}\}$, there exists some vector $\beta^* = (\beta_1^*, \dots, \beta_{n+1}^*) \in [\alpha - 1, \alpha]^{n+1}$ such that

$$0 = \frac{1}{n+1} \sum_{i=1}^{n+1} 1\{S_i^{(X_{n+1}, Y_{n+1})} \neq \rho_n^{(X_{n+1}, Y_{n+1})}(X_i)\} \left[ \theta \circ f_n^{(X_{n+1}, Y_{n+1})}(X_i)\{\alpha - 1(\rho_n^{(X_{n+1}, Y_{n+1})}(X_i) < S_i^{(X_{n+1}, Y_{n+1})})\} \right]$$

$$+ \frac{1}{n+1} \sum_{i=1}^{n+1} 1\{S_i^{(X_{n+1}, Y_{n+1})} = \rho_n^{(X_{n+1}, Y_{n+1})}(X_i)\} \left[ \beta_i^*(\theta \circ f_n^{(X_{n+1}, Y_{n+1})})(X_i) \right]$$

The above display can be rewritten as:

$$\frac{1}{n+1} \sum_{i=1}^{n+1} \left[ \theta \circ f_n^{(X_{n+1}, Y_{n+1})}(X_i)\{\alpha - 1(\rho_n^{(X_{n+1}, Y_{n+1})}(X_i) < S_i^{(X_{n+1}, Y_{n+1})})\} \right]$$

$$= \frac{1}{n+1} \sum_{i=1}^{n+1} 1\{S_i^{(X_{n+1}, Y_{n+1})} = \rho_n^{(X_{n+1}, Y_{n+1})}(X_i)\} \left[ (1 - \beta_i^*)(\theta \circ f_n^{(X_{n+1}, Y_{n+1})})(X_i) \right]. \tag{7}$$

Now, observe that the collection of random variables

$$\left\{ (f_n^{(X_{n+1}, Y_{n+1})}(X_i), S_i^{(X_{n+1}, Y_{n+1})}, \rho_n^{(X_{n+1}, Y_{n+1})}(X_i)) : i \in [n+1] \right\}$$

are exchangeable, since $\{((X_i, Y_i) : i \in [n+1]\}$ are exchangeable by C1 and the functions $f_n^{(X_{n+1}, Y_{n+1})}(\cdot)$ and $\rho_n^{(X_{n+1}, Y_{n+1})}(\cdot)$ are unchanged under permutations of the training data

$\{(X_i, Y_i)\}_{i \in [n+1]}$. Thus, the expectation of the left-hand side of Equation (7) can be expressed as:

$$\mathbb{E}\left[\frac{1}{n+1}\sum_{i=1}^{n+1}\left[\theta \circ f_n^{(X_{n+1},Y_{n+1})}(X_i)\{\alpha - 1(\rho_n^{(X_{n+1},Y_{n+1})}(X_i) < S_i^{(X_{n+1},Y_{n+1})})\}\right]\right]$$

$$= \frac{1}{n+1}\sum_{i=1}^{n+1}\mathbb{E}\left[\theta \circ f_n^{(X_{n+1},Y_{n+1})}(X_i)\{\alpha - 1(\rho_n^{(X_{n+1},Y_{n+1})}(X_i) < S_i^{(X_{n+1},Y_{n+1})})\}\right]$$

$$= \mathbb{E}\left[\theta \circ f_n^{(X_{n+1},Y_{n+1})}(X_{n+1})\{\alpha - 1(\rho_n^{(X_{n+1},Y_{n+1})}(X_{n+1}) < S_{n+1}^{(X_{n+1},Y_{n+1})})\}\right],$$

where the final inequality follows from exchangeability. Combining this with Equation (7), we find

$$\mathbb{E}\left[\theta \circ f_n^{(X_{n+1},Y_{n+1})}(X_{n+1})\{\alpha - 1(\rho_n^{(X_{n+1},Y_{n+1})}(X_{n+1}) < S_{n+1}^{(X_{n+1},Y_{n+1})})\}\right] \qquad (8)$$

$$= \mathbb{E}\left[1\{S_i^{(X_{n+1},Y_{n+1})} = \rho_n^{(X_{n+1},Y_{n+1})}(X_i)\}\left[(1 - \beta_i^*)(\theta \circ f_n^{(X_{n+1},Y_{n+1})})(X_i)\right]\right] \qquad (9)$$

**Lower bound on coverage:** We first obtain the lower coverage bound in the theorem statement. Note, for any nonnegative $\theta : \mathbb{R} \to \mathbb{R}$, that (9) implies:

$$\mathbb{E}\left[\theta \circ f_n^{(X_{n+1},Y_{n+1})}(X_{n+1})\{\alpha - 1(\rho_n^{(X_{n+1},Y_{n+1})}(X_{n+1}) < S_{n+1}^{(X_{n+1},Y_{n+1})})\}\right] \geq 0.$$

This inequality holds since $(1 - \beta_i^*) \geq 0$ and $(\theta \circ f_n^{(X_{n+1},Y_{n+1})})(X_i) \geq 0$ almost surely, for each $i \in [n+1]$.

By the law of iterated expectations, we then have

$$\mathbb{E}\left[\theta \circ f_n^{(X_{n+1},Y_{n+1})}(X_{n+1})\left\{\alpha - \mathbb{P}\left(\rho_n^{(X_{n+1},Y_{n+1})}(X_{n+1}) < S_{n+1}^{(X_{n+1},Y_{n+1})} \mid f_n^{(X_{n+1},Y_{n+1})}(X_{n+1})\right)\right\}\right] \geq 0.$$

Taking $\theta : \mathbb{R} \to \mathbb{R}$ as a nonnegative map that almost surely satisfies

$$\theta \circ f_n^{(X_{n+1},Y_{n+1})}(X_{n+1}) = 1\left\{\alpha \leq \mathbb{P}\left(\rho_n^{(X_{n+1},Y_{n+1})}(X_{n+1}) < S_{n+1}^{(X_{n+1},Y_{n+1})} \mid f_n^{(X_{n+1},Y_{n+1})}(X_{n+1})\right)\right\},$$

we find

$$-\mathbb{E}\left[\left\{\alpha - \mathbb{P}\left(\rho_n^{(X_{n+1},Y_{n+1})}(X_{n+1}) < S_{n+1}^{(X_{n+1},Y_{n+1})} \mid f_n^{(X_{n+1},Y_{n+1})}(X_{n+1})\right)\right\}_-\right] \geq 0,$$

where the map $t \mapsto \{t\}_- := |t|1(t \leq 0)$ extracts the negative part of its input. Multiplying both sides of the previous inequality by $-1$, we obtain

$$0 \leq \mathbb{E}\left[\left\{\alpha - \mathbb{P}\left(\rho_n^{(X_{n+1},Y_{n+1})}(X_{n+1}) < S_{n+1}^{(X_{n+1},Y_{n+1})} \mid f_n^{(X_{n+1},Y_{n+1})}(X_{n+1})\right)\right\}_-\right] \leq 0.$$

We conclude that the negative part of $\left\{\alpha - \mathbb{P}\left(\rho_n^{(X_{n+1},Y_{n+1})}(X_{n+1}) < S_{n+1}^{(X_{n+1},Y_{n+1})} \mid f_n^{(X_{n+1},Y_{n+1})}(X_{n+1})\right)\right\}$ is almost surely zero. Thus, it must be almost surely true that

$$\alpha \geq \mathbb{P}\left(\rho_n^{(X_{n+1},Y_{n+1})}(X_{n+1}) < S_{n+1}^{(X_{n+1},Y_{n+1})} \mid f_n^{(X_{n+1},Y_{n+1})}(X_{n+1})\right).$$

Note that the event $Y_{n+1} \in \widehat{C}_{n+1}(X_{n+1}) = \{y \in \mathcal{Y} : S_{n+1}^{(X_{n+1},y)} \leq \rho_n^{(X_{n+1},y)}(X_{n+1})\}$ occurs if, and only if, $\rho_n^{(X_{n+1},Y_{n+1})}(X_{n+1}) \geq S_{n+1}^{(X_{n+1},Y_{n+1})}$. As a result, we obtain the desired lower coverage bound:

$$1 - \alpha \leq \mathbb{P}\left(Y_{n+1} \in \widehat{C}_{n+1}(X_{n+1}) \mid f_n^{(X_{n+1},Y_{n+1})}(X_{n+1})\right).$$

**Deviation bound for the coverage:** We now bound the deviation of the coverage of SC-CP from the lower bound. Note, for any $f : \mathbb{R} \to \mathbb{R}$, that (9) implies:

$$\mathbb{E}\left[\theta \circ f_n^{(X_{n+1},Y_{n+1})}(X_{n+1})\{\alpha - 1(\rho_n^{(X_{n+1},Y_{n+1})}(X_{n+1}) < S_{n+1}^{(X_{n+1},Y_{n+1})})\}\right]$$

$$\leq \left|\mathbb{E}\left[1\{S_i^{(X_{n+1},Y_{n+1})} = \rho_n^{(X_{n+1},Y_{n+1})}(X_i)\}\left[(1 - \beta_i^*)(\theta \circ f_n^{(X_{n+1},Y_{n+1})})(X_i)\right]\right]\right|. \qquad (10)$$

Using that $(1 - \beta_i^*) \in [0, 1]$ and exchangeability, we can bound the right-hand side of the above as

$$
\left| \mathbb{E} \left[ \frac{1}{n+1} \sum_{i=1}^{n+1} 1\{S_i^{(X_{n+1}, Y_{n+1})} = \rho_n^{(X_{n+1}, Y_{n+1})}(X_i)\} \left[ (1 - \beta_i^*)(\theta \circ f_n^{(X_{n+1}, Y_{n+1})})(X_i) \right] \right] \right|
$$

$$
\leq \frac{1}{n+1} \mathbb{E} \left[ \left\{ \max_{i \in [n+1]} \left| (\theta \circ f_n^{(X_{n+1}, Y_{n+1})})(X_i) \right| \right\} \sum_{i=1}^{n+1} 1\{S_i^{(X_{n+1}, Y_{n+1})} = \rho_n^{(X_{n+1}, Y_{n+1})}(X_i)\} \right],
$$

Next, since there are no ties by C3, the event $1\{S_i^{(X_{n+1}, Y_{n+1})} = \rho_n^{(X_{n+1}, Y_{n+1})}(X_i)\}$ for some index $i \in [n+1]$ can only occur once per piecewise constant segment of $\rho_n^{(X_{n+1}, Y_{n+1})}$, since, otherwise, $S_i^{(X_{n+1}, Y_{n+1})} = \rho_n^{(X_{n+1}, Y_{n+1})}(X_i) = \rho_n^{(X_{n+1}, Y_{n+1})}(X_j) = S_j^{(X_{n+1}, Y_{n+1})}$ for some $i \neq j$. However, $\rho_n^{(X_{n+1}, Y_{n+1})}$ is a transformation of $f_n^{(X_{n+1}, Y_{n+1})}$ and, therefore, has the same number of constant segments as $f_n^{(X_{n+1}, Y_{n+1})}$. Thus, it holds that

$$
\sum_{i=1}^{n+1} 1\{S_i^{(X_{n+1}, Y_{n+1})} = \rho_n^{(X_{n+1}, Y_{n+1})}(X_i)\} \leq N^{(X_{n+1}, Y_{n+1})},
$$

where $N^{(X_{n+1}, Y_{n+1})}$ is the (random) number of constant segments of $f_n^{(X_{n+1}, Y_{n+1})}$. This implies that

$$
\frac{1}{n+1} \mathbb{E} \left[ \max_{i \in [n+1]} |(\theta \circ f_n^{(X_{n+1}, Y_{n+1})})(X_i)| \sum_{i=1}^{n+1} 1\{S_i^{(X_{n+1}, Y_{n+1})} = \rho_n^{(X_{n+1}, Y_{n+1})}(X_i)\} \right]
$$

$$
\leq \frac{1}{n+1} \mathbb{E} \left[ N^{(X_{n+1}, Y_{n+1})} \max_{i \in [n+1]} |(\theta \circ f_n^{(X_{n+1}, Y_{n+1})})(X_i)| \right].
$$

Combining this bound with (10), we find

$$
\mathbb{E} \left[ \theta \circ f_n^{(X_{n+1}, Y_{n+1})}(X_{n+1})\{\alpha - 1(\rho_n^{(X_{n+1}, Y_{n+1})}(X_{n+1}) < S_{n+1}^{(X_{n+1}, Y_{n+1})})\} \right]
$$

$$
\leq \frac{1}{n+1} \mathbb{E} \left[ N^{(X_{n+1}, Y_{n+1})} \max_{i \in [n+1]} |(\theta \circ f_n^{(X_{n+1}, Y_{n+1})})(X_i)| \right].
$$

By the law of iterated expectations, we then have

$$
\mathbb{E} \left[ \theta \circ f_n^{(X_{n+1}, Y_{n+1})}(X_{n+1}) \left\{ \alpha - \mathbb{P} \left( \rho_n^{(X_{n+1}, Y_{n+1})}(X_{n+1}) < S_{n+1}^{(X_{n+1}, Y_{n+1})} \mid f_n^{(X_{n+1}, Y_{n+1})}(X_{n+1}) \right) \right\} \right]
$$

$$
\leq \frac{1}{n+1} \mathbb{E} \left[ N^{(X_{n+1}, Y_{n+1})} \max_{i \in [n+1]} |(\theta \circ f_n^{(X_{n+1}, Y_{n+1})})(X_i)| \right].
$$

Next, let $\mathcal{V} \subset \mathbb{R}$ denote the support of the random variable $f_n^{(X_{n+1}, Y_{n+1})}(X_{n+1})$. Then, taking $\theta$ to be $t \mapsto 1(t \in \mathcal{V})\text{sign}\left\{ \alpha - \mathbb{P} \left( \rho_n^{(X_{n+1}, Y_{n+1})}(X_{n+1}) < S_{n+1}^{(X_{n+1}, Y_{n+1})} \mid f_n^{(X_{n+1}, Y_{n+1})}(X_{n+1}) = t \right) \right\}$, which falls almost surely in $\{-1, 1\}$, we obtain the mean absolute error bound:

$$
\mathbb{E} \left| \alpha - \mathbb{P} \left( \rho_n^{(X_{n+1}, Y_{n+1})}(X_{n+1}) < S_{n+1}^{(X_{n+1}, Y_{n+1})} \mid f_n^{(X_{n+1}, Y_{n+1})}(X_{n+1}) \right) \right| \leq \frac{1}{n+1} \mathbb{E} \left[ N^{(X_{n+1}, Y_{n+1})} \right].
$$

Since the event $Y_{n+1} \notin \widehat{C}_{n+1}(X_{n+1}) = \{y \in \mathcal{Y} : S_{n+1}^{(X_{n+1}, y)} \leq \rho_n^{(X_{n+1}, y)}(X_{n+1})\}$ occurs if, and only if, $\rho_n^{(X_{n+1}, Y_{n+1})}(X_{n+1}) < S_{n+1}^{(X_{n+1}, Y_{n+1})}$, we conclude that

$$
\mathbb{E} \left| \alpha - \mathbb{P} \left( Y_{n+1} \notin \widehat{C}_{n+1}(X_{n+1}) \mid f_n^{(X_{n+1}, Y_{n+1})}(X_{n+1}) \right) \right| \leq \frac{\mathbb{E}[N^{(X_{n+1}, Y_{n+1})}]}{n+1}.
$$

Under C4, $\mathbb{E}[N^{(X_{n+1}, Y_{n+1})}] \leq n^{1/3} \text{polylog } n$, such that

$$
\mathbb{E} \left| \alpha - \mathbb{P} \left( Y_{n+1} \notin \widehat{C}_{n+1}(X_{n+1}) \mid f_n^{(X_{n+1}, Y_{n+1})}(X_{n+1}) \right) \right| \leq \frac{n^{1/3} \text{polylog } n}{n+1} \leq \frac{\text{polylog } n}{n^{2/3}},
$$

as desired.

$\square$

*Proof of Theorem 4.3.* Let $P_{n+1}$ denote the empirical distribution of $\{(X_i, Y_i)\}_{i=1}^{n+1}$ and let $P_n$ denote the empirical distribution of $\{(X_i, Y_i)\}_{i=1}^n$. For any function $g : \mathcal{X} \times \mathcal{Y} \to \mathbb{R}$: we use the following empirical process notation: $Pg := \int g(x,y) dP(x,y)$, $P_{n+1}g := \int g(x,y) dP_{n+1}(x,y)$, and $P_n g := \int g(x,y) dP_n(x,y)$.

Define the risk functions $R_n^{(x,y)}(\theta) := \frac{1}{n+1} \sum_{i=1}^n \{S_\theta(X_i, Y_i)\}^2 + \frac{1}{n+1}\{S_\theta(x,y)\}^2$, $R_{n+1}(\theta) := \frac{1}{n+1} \sum_{i=1}^{n+1} \{S_\theta(X_i, Y_i)\}^2$, and $R_0(\theta) := \int \{S_\theta(x,y)\}^2 dP(x,y)$. Moreover, define the risk minimizers as $\theta_n^{(x,y)} := \operatorname{argmin}_{\theta \in \Theta_{iso}} R_n^{(x,y)}(f)$ and $\theta_0 := \operatorname{argmin}_{\theta \in \Theta_{iso}} R_0(\theta)$. Observe that $R_n^{(x,y)}(\theta_n^{(x,y)}) - R_n^{(x,y)}(\theta_0) \le 0$ since $f_n$ minimizes $R_n$ over $\Theta_{iso}$. Using this inequality, it follows that

$$
\begin{aligned}
R_0(\theta_n^{(x,y)}) - R_0(\theta_0) &= R_0(\theta_n^{(x,y)}) - R_n^{(x,y)}(\theta_n^{(x,y)}) \\
&\quad + R_n^{(x,y)}(\theta_n^{(x,y)}) - R_n^{(x,y)}(\theta_0) + R_n^{(x,y)}(\theta_0) - R_0(\theta_0) \\
&\le R_0(\theta_n^{(x,y)}) - R_n^{(x,y)}(\theta_n^{(x,y)}) - \left\{ R_n^{(x,y)}(\theta_0) - R_0(\theta_0) \right\} \\
&\le R_0(\theta_n^{(x,y)}) - R_{n+1}(\theta_n^{(x,y)}) - \{R_{n+1}(\theta_0) - R_0(\theta_0)\} \\
&\quad + R_n^{(x,y)}(\theta_n^{(x,y)}) - R_{n+1}(\theta_n^{(x,y)}) - \left\{ R_n^{(x,y)}(\theta_0) - R_{n+1}(\theta_0) \right\}.
\end{aligned}
$$

The first term on the right-hand side of the above display can written as

$$
R_0(\theta_n^{(x,y)}) - R_{n+1}(\theta_n^{(x,y)}) - \{R_{n+1}(\theta_0) - R_0(\theta_0)\} = (P_{n+1} - P)\left[ \{S_{\theta_n^{(x,y)}}\}^2 - \{S_{\theta_0}\}^2 \right].
$$

We now bound the second term, $R_n^{(x,y)}(\theta_n^{(x,y)}) - R_{n+1}(\theta_n^{(x,y)}) - \left\{ R_n^{(x,y)}(\theta_0) - R_{n+1}(\theta_0) \right\}$. For any $\theta \in \Theta_{iso}$, observe that

$$
R_n^{(x,y)}(\theta) - R_{n+1}(\theta) = \frac{1}{n+1} \left[ \{y - \theta \circ f(x)\}^2 - \{Y_{n+1} - \theta \circ f(X_{n+1})\}^2 \right].
$$

We know that $\theta_n^{(x,y)}$ and $\theta_0$, being defined via isotonic regression, are uniformly bounded by $B := \sup_{y \in \mathcal{Y}} |y|$, which is finite by C6. Therefore,

$$
\left| R_n^{(x,y)}(\theta_n^{(x,y)}) - R_{n+1}(\theta_n^{(x,y)}) - \left\{ R_n^{(x,y)}(\theta_0) - R_{n+1}(\theta_0) \right\} \right| \le \frac{8B^2}{n+1} = O(n^{-1}).
$$

Combining the previous displays, we obtain the excess risk bound

$$
R_0(\theta_n^{(x,y)}) - R_0(\theta_0) \le (P_{n+1} - P)\left[ \{S_{\theta_n^{(x,y)}}\}^2 - \{S_{\theta_0}\}^2 \right] + O(n^{-1}). \tag{11}
$$

Next, we claim that $R_0(\theta_n^{(x,y)}) - R_0(\theta_0) \ge \|(\theta_n^{(x,y)} \circ f) - (\theta_0 \circ f)\|_P^2$. To show this, expanding the squares, note, pointwise for each $x \in \mathcal{X}$ and $y \in \mathcal{Y}$, that

$$
\begin{aligned}
\{S_{\theta_n^{(x,y)}}(x,y)\}^2 - \{S_{\theta_0}(x,y)\}^2 &= \{\theta_n^{(x,y)} \circ f(x)\}^2 - \{\theta_0 \circ f(x)\}^2 - 2y\left\{ (\theta_n^{(x,y)} \circ f)(x) - (\theta_0 \circ f)(x) \right\} \\
&= \left\{ (\theta_n^{(x,y)} \circ f)(x) + (\theta_0 \circ f)(x) - 2y \right\} \left\{ (\theta_n^{(x,y)} \circ f)(x) - (\theta_0 \circ f)(x) \right\}.
\end{aligned}
$$

Consequently,

$$
R_0(\theta_n^{(x,y)}) - R_0(\theta_0) = \int \left\{ (\theta_n^{(x,y)} \circ f)(x) + (\theta_0 \circ f)(x) - 2y \right\} \left\{ (\theta_n^{(x,y)} \circ f)(x) - (\theta_0 \circ f)(x) \right\} dP(x). \tag{12}
$$

The class $\Theta_{iso}$ consists of all isotonic functions and is, therefore, a convex space. Thus, the first-order derivative equations defining the population minimizer $\theta_0$ imply that

$$
\int \left\{ \theta_n^{(x,y)} \circ f(x) - \theta_0 \circ f(x) \right\} \{ y - \theta_0 \circ f(x) \} dP(x,y) \le 0. \tag{13}
$$

Combining (12) and (13), we find

$$R_0(\theta_n^{(x,y)}) - R_0(\theta_0) = \int \left\{ (\theta_n^{(x,y)} \circ f)(x) - (\theta_0 \circ f)(x) \right\}^2 dP(x)$$

$$+ 2 \int \left\{ (\theta_0 \circ f)(x) - y \right\} \left\{ (\theta_n^{(x,y)} \circ f)(x) - (\theta_0 \circ f)(x) \right\} dP(x)$$

$$\geq \int \left\{ (\theta_n^{(x,y)} \circ f)(x) - (\theta_0 \circ f)(x) \right\}^2 dP(x),$$

as desired. Combining this lower bound with (11), we obtain the inequality

$$\int \left\{ (\theta_n^{(x,y)} \circ f)(x) - (\theta_0 \circ f)(x) \right\}^2 dP(x) \leq R_0(\theta_n^{(x,y)}) - R_0(\theta_0) \leq (P_n - P) \left[ \{ S_{\theta_n^{(x,y)}} \}^2 - \{ S_{\theta_0} \}^2 \right] + O(1/n)$$

$$(14)$$

Define $\delta_n := \sqrt{\int \left\{ (\theta_n^{(x,y)} \circ f)(x) - (\theta_0 \circ f)(x) \right\}^2 dP(x)}$, the bound $B := \sup_{y \in \mathcal{Y}} |y|$, and the function class,

$$\Theta_{1,n} := \{ (x,y) \mapsto \{ (\theta_1 + \theta_2) \circ f - 2y \} \{ (\theta_1 - \theta_2) \circ f \} \}.$$

Using this notation, (14) implies

$$\delta_n^2 \leq \sup_{\theta_1, \theta_2 \in \Theta_{iso} : \|\theta_1 - \theta_2\| \leq \delta_n} \int \{ (\theta_1 + \theta_2) \circ f(x) - 2y \} \{ (\theta_1 - \theta_2) \circ f(x) \} d(P_n - P)(x,y) + O(1/n)$$

$$\leq \sup_{h \in \Theta_{1,n} : \|h\| \leq 4B\delta_n} (P_n - P)h + O(1/n)$$

Using the above inequality and C5, we will use an argument similar to the proof of Theorem 3 in [52] to establish that $\delta_n = O_p(n^{-1/3})$. This rate then implies the result of the theorem. To see this, note, by the reverse triangle inequality,

$$\left| S_n^{(x,y)}(y', x') - S_0(x', y') \right| = \left| |y' - \theta_n^{(x,y)}(x')| - |y' - \theta_0(x')| \right|$$

$$\leq \left| \theta_0(x') - \theta_n^{(x,y)}(x') \right|.$$

Squaring and integrating the left- and right-hand sides, we find

$$\int \left| S_n^{(x,y)}(y', x') - S_0(x', y') \right|^2 dP(x', y') \leq \int \left| \theta_0(x') - \theta_n^{(x,y)}(x') \right|^2 dP(x') = \delta_n^2,$$

as desired.

We now establish that $\delta_n = O_p(n^{-1/3})$. For a function class $\mathcal{F}$, let $N(\epsilon, \mathcal{F}, L_2(P))$ denote the $\epsilon$−covering number [53] of $\mathcal{F}$ and define the uniform entropy integral of $\mathcal{F}$ by

$$\mathcal{J}(\delta, \mathcal{F}) := \int_0^\delta \sup_Q \sqrt{\log N(\epsilon, \mathcal{F}, L_2(Q))} \, d\epsilon \,,$$

where the supremum is taken over all discrete probability distributions $Q$. We note that

$$\mathcal{J}(\delta, \Theta_{1,n}) = \int_0^\delta \sup_Q \sqrt{N(\varepsilon, \Theta_{1,n}, \| \cdot \|_Q)} \, d\varepsilon = \int_0^\delta \sup_Q \sqrt{N(\varepsilon, \Theta_{iso}, \| \cdot \|_{Q \circ f^{-1}})} \, d\varepsilon = \mathcal{J}(\delta, \Theta_{iso}) \,,$$

where $Q \circ f^{-1}$ is the push-forward probability measure for the random variable $f(W)$. Additionally, the covering number bound for bounded monotone functions given in Theorem 2.7.5 of [53] implies that

$$\mathcal{J}(\delta, \Theta_{1,n}) = \mathcal{J}(\delta, \Theta_{iso}) \lesssim \sqrt{\delta}.$$

Recall that $f$ is obtained from an external dataset, say $\mathcal{E}_n$, independent of the calibration data. Noting that $f$ is deterministic conditional on a training dataset $\mathcal{E}_n$. Applying Theorem 2.1 of [54] conditional on $\mathcal{E}_n$, we obtain, for any $\delta > 0$, that

$$E \left[ \sup_{h \in \Theta_{1,n} : \|h\| \leq 4B\delta} (P_n - P)h \,\middle|\, \mathcal{E}_n \right] \lesssim n^{-1/2} \mathcal{J}(\delta, \Theta_{1,n}) \left( 1 + \frac{\mathcal{J}(\delta, \Theta_{1,n})}{\sqrt{n}\delta^2} \right).$$

$$\lesssim n^{-1/2} \mathcal{J}(\delta, \Theta_{iso}) \left( 1 + \frac{\mathcal{J}(\delta, \Theta_{iso})}{\sqrt{n}\delta^2} \right).$$

Noting that the right-hand side of the above bound is deterministic, we conclude that

$$E\left[\sup_{h\in\Theta_{1,n}:\|h\|\leq 4B\delta}(P_n-P)h\right] \lesssim n^{-1/2}\mathcal{J}(\delta,\Theta_{iso})\left(1+\frac{\mathcal{J}(\delta,\Theta_{iso})}{\sqrt{n}\delta^2}\right).$$

We use the so-called "peeling" argument [53] to obtain our bound for $\delta_n$. Note

$$
\begin{aligned}
P\left(\delta_n^2 \geq n^{-2/3}2^M\right) &= \sum_{m=M}^{\infty} P\left(2^{m+1} \geq n^{2/3}\delta_n^2 \geq 2^m\right) \\
&= \sum_{m=M}^{\infty} P\left(2^{m+1} \geq n^{2/3}\delta_n^2 \geq 2^m, \delta_n^2 \leq \sup_{h\in\Theta_{1,n}:\|h\|\leq 4B\delta_n}(P_n-P)h + O(1/n)\right) \\
&= \sum_{m=M}^{\infty} P\left(2^{m+1} \geq n^{2/3}\delta_n^2 \geq 2^m, 2^{2m}n^{-2/3} \leq \sup_{h\in\Theta_{1,n}:\|h\|\leq 4B2^{m+1}n^{-1/3}}(P_n-P)h + O(1/n)\right) \\
&\leq \sum_{m=M}^{\infty} P\left(2^{2m}n^{-2/3} \leq \sup_{h\in\Theta_{1,n}:\|h\|\leq 4B2^{m+1}n^{-1/3}}(P_n-P)h + O(1/n)\right) \\
&\leq \sum_{m=M}^{\infty} \frac{E\left[\sup_{h\in\Theta_{1,n}:\|h\|\leq 4B2^{m+1}n^{-1/3}}(P_n-P)h\right] + O(1/n)}{2^{2m}n^{-2/3}} \\
&\leq \sum_{m=M}^{\infty} \frac{\mathcal{J}(2^{m+1}n^{-1/3},\Theta_{iso})\left(1+\frac{\mathcal{J}(2^{2m+2}n^{-2/3},\Theta_{iso})}{\sqrt{n}2^{m+1}n^{-1/3}}\right)}{\sqrt{n}2^{2m}n^{-2/3}} + \sum_{m=M}^{\infty}\frac{O(1/n)}{2^{2m}n^{-2/3}} \\
&\leq \sum_{m=M}^{\infty} \frac{2^{(m+1)/2}n^{-1/6}}{2^{2m}n^{-1/6}} + \sum_{m=M}^{\infty}\frac{o(1)}{2^{2m}} \\
&\lesssim \sum_{m=M}^{\infty} \frac{2^{(m+1)/2}}{2^{2m}}.
\end{aligned}
$$

Since $\sum_{m=1}^{\infty}\frac{2^{(m+1)/2}}{2^{2m}} < \infty$, we have that $\sum_{m=M}^{\infty}\frac{2^{(m+1)/2}}{2^{2m}} \to 0$ as $M \to \infty$. Therefore, for all $\varepsilon > 0$, we can choose $M > 0$ large enough so that

$$P\left(\delta_n^2 \geq n^{-2/3}2^M\right) \leq \varepsilon.$$

We conclude that $\delta_n = O_p(n^{-1/3})$ as desired.

$\square$

