# OpenReview forum: "Self-Calibrating Conformal Prediction"
_NeurIPS.cc/2024/Conference — NeurIPS 2024 poster_

### Official Review · Reviewer_qyB7 · 2024-06-26

**Soundness:** 2
**Presentation:** 2
**Contribution:** 3
**Rating:** 5
**Confidence:** 3

**Summary:**

The authors propose a new scheme to obtain prediction intervals that are valid conditioning on the model output. The approach leverages the notion of Perfectly Calibrated Point Predictors and an extension of Venn-Abers calibration to the regression setup.

**Strengths:**

- I like the interpretation of $f(X)$ as a "scalar dimension reduction".
- Generalizing self-consistency to prediction sets may inspire future investigation in the broader CP community.
- The idea of leveraging Venn-Abers calibration to the regression setup is a valuable contribution.

**Weaknesses:**

- The authors do not explain, in an intuitive way, why conditioning on $f(X)$ is a good approximation of conditioning on $X$. They could describe their 1D dimensional-reduction argument better, possibly with a concrete example. Imagine $(X, Y)\in {\mathbb R}$ and $Y \sim ({\bf 1}(X<0) + 10 {\bf 1}(X<0)) {\cal N}(0, 1)$. A perfect point-predictor is $f(X) = {\rm E}(Y|X) =0$ for all $X$. In this case, conditioning on $X$ or $f(X)$ does not look equivalent.
- The description of Venn-Abers calibration can be improved. I have not fully understood the meaning of these two sentences [1][2].
- The authors may explain more intuitively why they need a Perfectly Calibrated Point Prediction to compute a Self Calibrated Prediction Interval.
- The proposed model is only compared with Mondrian CP.

[1] *Venn-Abers calibration accounts for overfitting by widening the range of the multi-prediction in such scenarios, thus indicating greater uncertainty in the value of the perfectly calibrated point prediction.*

[2] *Each point prediction in the set enjoys the same large-sample calibration guarantees for isotonic calibration.*

**Questions:**

- Intuitively, how does conditioning on $f(X)$ help? How does this avoid the *curse of dimensionality*? What is the difference compared with conditioning on $Y$?
- Is $1({Y \in C})$ the indicator of ${Y \in C}$? Why is $f$ called a *covariate shift* in (3)? How is this the same as saying that $f(X)$ is the model output?
- Has prediction-conditional validity been used before?
- How is $f^{X, y}$ defined? Should $f_{n+1} =  ( f_n^{X_{n+1}, y}(X_{n+1}), y \in {\cal Y} ) $ be interpreted as a recursive definition? How does $f_{n+1}$ depend on $x$?
- Would it be possible to compute the marginal prediction band centered on the Venn-Abers calibrated predictor (the black lines in Figure 2, if I understand the plots correctly) instead of the original one?

**Limitations:**

The authors should have said why they do not compare with other approximations of context-conditional validity, e.g. Conformal Quantile Regression, Ref 22 in the paper, or the Error Reweighting method [3].


[3]
Papadopoulos, Harris, Alex Gammerman, and Volodya Vovk. *Normalized nonconformity measures for regression conformal prediction.* Proceedings of the IASTED International Conference on Artificial Intelligence and Applications (AIA 2008). 2008.

---

> ### Author Rebuttal · Authors · 2024-08-07
>
> ## Response to weaknesses
>
> 1.  We have added clarification on when our proposed self-calibration objective can approximate feature-conditional validity. Our self-calibration objective aims to provide a relaxation of feature-conditional validity that is feasible in finite samples (Section 2.3). A key benefit is that it involves conditioning on a one-dimensional variable, thus avoiding the curse of dimensionality (Section 2.2). Self-calibration is not a replacement for feature-conditional validity. Prediction-conditional validity can approximate it when the outcome's heteroscedasticity/variance is a function of its conditional mean. Appendix B.3 experimentally confirms this using synthetic data, and Section 5 provides additional evidence for this using real data that appears to have a mean-variance relationship.▍
>
> 2. We have redrafted the description for clarity and provided the algorithm before describing its qualitative properties. The revised sentences include:
>
>     1. "Isotonic calibration can overfit, leading to poorly calibrated predictions. When this occurs, the Venn-Abers set prediction widens, reflecting greater (epistemic) uncertainty in the perfectly calibrated point prediction within the set."
>     2. "The Venn-Abers calibration set is guaranteed to contain at least one perfectly calibrated point prediction in finite samples, and each prediction, being obtained via isotonic calibration, still enjoys the same large-sample calibration guarantees as isotonic calibration."
>
> 3. We clarified the role of calibration in the self-calibration objective, which aims to obtain prediction intervals that are centered around calibrated point predictions and provide valid coverage conditional on the calibrated point prediction. Point calibration ensures that the prediction interval is centered around a conditionally unbiased point prediction. Prediction-conditional validity is only attainable in finite samples for predictors with discrete outputs. Venn-Abers calibration discretizes the predictor, enabling prediction-conditional validity while mitigating the loss in predictive power due to discretization.
>
> 4.   Along with Mondrian-CP, we also included the kernel smoothing approach of Gibbs et al. (2023). Our baselines are sufficient to illustrate the benefit of our approach combining point calibration and prediction-conditional validity. The baselines for prediction-conditional involve estimating a one-dimensional quantile function, where Mondrian-CP uses histogram regression and Gibbs et al. use kernel smoothing. Kernel smoothing and histogram regression are minimax optimal for 1D functions under weak assumptions. Also see our response to Limitation #1.
>
> ## Response to Questions
>
> 1. See also our response to Weakness 1. On the difference between conditioning on $ Y $ and $ f(X) $:  $ Y $ is typically a noisy signal $ f_{true}(X) + \varepsilon $. Thus, two contexts $ X_1 $ and $ X_2 $ with $Y_1 = Y_2$ can have very different signals $ f_{true}(X_1) $ and $ f_{true}(X_2) $. Outcome conditional validity is primarily useful in classification settings where the outcome $ Y $ is a ground-truth label and thus not subject to noise. We have added this discussion to related work.
>
> 2.  Yes, it is the set indicator. The covariate shift (CS) terminology for the multicalibration objective in (3) is from Gibbs et al. (2023). In equation (3), $ f $ is not the predictor but an arbitrary element of $\mathcal{F}$. Thank you for pointing out this confusion in notation. We now denote the CS $ f $ in (3) by $ h $. The CS terminology arises in (3) because if $ h $ is a density ratio between a source and target distribution then multicalibration with respect to $ h $ in the source population implies marginal coverage with respect to the target distribution as well.
>
> 3. Prediction-conditional validity as a formal objective has not been proposed before. Some works have used bins based on predictions as an application of Mondrian CP to provide a coarse form of prediction-conditional validity. Our contributions include extending Venn-Abers calibration to regression and proposing the self-calibration objective and our solution, combining prediction-conditional validity with data-adaptive output discretization with the calibration of model outputs.
>
> 4.   $f_n^{(x,y)} $ is defined in Algorithm 1. The subscript $ n $ indicates that the model depends on the calibration data and the superscript $(x,y)$ indicates that the model also depends on the context $ x $ and the imputed outcome $ y $. In our notation for $ f_{n+1} $, we suppressed the dependence on the context $ x $. To avoid confusion and make the dependence on $ x $ explicit, we have now changed the notation from $ f_{n+1} $ to $ f_{n,x} $. We fixed a typo on line 153, where $ f_{n+1}(x) $ is now $ f_{n+1}(X_{n+1}) $.
>
> 5.  It is not possible to compute the marginal bands obtained using split CP around the calibrated predictor without sacrificing finite-sample validity, as Venn-Abers calibration is performed using the same data used to construct CP bands.
>
> ## Response to limitations
>
> 1. We have clarified our choice of baselines, focusing on those targeting prediction-conditional validity as that is part of our self-calibration objective. We will add Conformal Quantile Regression as a feature-conditional baseline in the revised experiments. For direct comparison with our method, our baselines employed the commonly-used absolute residual error conformity score. While different conformity scores can improve feature-conditional validity, our primary goal is to show how calibrated point predictions and prediction-conditional validity enhance interval efficiency and adaptivity. Our method can be adapted to any conformity score, including the error reweighting score, to provide variance-adaptive prediction bands. We now discuss this in the conclusion and provide an explicit algorithm for the Error Reweighting modification in the Appendix.

---

> > ### Comment · Reviewer_qyB7 · 2024-08-08
> > **thank you for your answers**
> >
> > I am happy with the authors' explanations and will raise my score to 5.

---

### Official Review · Reviewer_QroB · 2024-07-09

**Soundness:** 3
**Presentation:** 3
**Contribution:** 2
**Rating:** 6
**Confidence:** 2

**Summary:**

The paper proposes a method that jointly calibrates point predictions and provides prediction intervals with valid coverage given the point predictions. This is performed by combining two existing post-hoc processing procedures, Venn-Abers calibration and conformal prediction. The analysis of the method provides guarantees as in CP methods and a convergence rate for the conformity scoring function. The method provides a calibrated model that is piecewise constant and adapts to outcome heteroscedasticity.

**Strengths:**

The paper is clearly written and provides a good overview of the background material on both CP and Venn-Abers calibration. The motivation for the methodology is sound. The main contribution of the paper is the theoretical analysis that combines existing results in a novel way. The algorithm is well-presented and appears relatively easy to implement.

**Weaknesses:**

One of the main disadvantages of the proposed approach is the computational complexity, as discussed in Sec. 3.4. Due to the high computational complexity, it may not offer significant advantage over other existing methods, e.g. the Mondrian CP. It is not immediately clear from the paper what is the advantage offered over existing methods.

**Questions:**

What is the meaning of the subscript of $f_{n+1}$ in the desiderata (line 90)?

The colour (blue) of the results in Fig. 2 is not helpful. Would it be better to highlight which methods perform best?

Is there a reason you are not comparing to the method in [42]? It seems to be the closes related method.

The example presented in the experiment section is interesting and helpful in explaining the methodology. However, could you provide other examples of datasets or problems where your methodology would offer significant advantage over existing methods? And when would it fail in comparison to existing methods?

Did your theoretical analysis offer any general insights about the class of problems that you are addressing in this work? Do you consider the theoretical analysis to be the main contribution of this work? Can the analysis be reused or easily adapted to other methods?

Minor:
The citations do not seem to be linked.

**Limitations:**

The paper contains a discussion of the limitations (in Sec. 3.4) as well as appendix B.3. However, I believe that the limitations discussed in the Paper Checklist should be moved to the main part of the paper as it would contribute to the exposition of the methodology.

---

> ### Author Rebuttal · Authors · 2024-08-07
>
> ## Weaknesses
>
> 1. **Weakness:** One of the main disadvantages of the proposed approach is the computational complexity... it may not offer significant advantage over other existing methods.
>
>    **Response:** While our method has greater computational complexity than the split-CP approach for marginal validity, it is faster compared to methods like full CP and the conditional split CP approach by Gibbs et al. (2023). It is scalable to large datasets, as isotonic regression can be computed with XGBoost. In our experiments with calibration datasets (n=10000-50000), the computational time ranged from 1 to 5 minutes. Given that training the initial/uncalibrated model can take much longer, we believe the computational complexity of our method is not a significant weakness.
>
>     Advantages of our method over existing methods are discussed in Section 4.1 and shown empirically in Section 5. Mondrian-CP approaches can only provide prediction-conditional validity and do not achieve our objective of self-calibration, which offers both calibrated point predictions and prediction-conditional validity. Another limitation of Mondrian-CP is the need for pre-specification of a binning scheme for the predictor f(·), introducing a trade-off between model performance and prediction interval width. In contrast, SC-CP data-adaptively discretizes the predictor f(·) using isotonic calibration, providing calibrated predictions, improved conformity scores, and self-calibrated intervals.
>
> ## Questions
>
> 1. What is the meaning of the subscript of $f_{n+1}$ in the desiderata (line 90)?
>
>
>    **Response:** The subscript indicates that $f_{n+1}$ is obtained by calibrating the original model $f$ using the calibration data $(X_i,Y_i): i \in [n]$ and the new context $X_{n+1}$. We have clarified in our objective how $f_{n+1}$ is obtained and explained this notation in the revised paper.
>
>
>
>
> 2. ... Would it be better to highlight which methods perform best?
>
>     **Response:** We have changed the coloring in the figure to highlight which method performs best.
>
>
>
> 3. "Is there a reason you are not comparing to the method in [42]?"
>
>     **Response:** We did not include the method of [42] because their aim is to construct cumulative distribution function (CDF) estimates for a continuous outcome with marginal calibration guarantees. Our objective is to construct prediction intervals centered around a calibrated point prediction with prediction-conditional coverage guarantees. Inverting the CDF estimates from [42] yields quantile estimates and prediction intervals. However, these intervals are not centered around a point prediction, don't use conformity scores, and do not ensure prediction-conditional validity. While both approaches use Venn-Abers calibration, we use it to calibrate the regression model and employ conformal prediction techniques to construct prediction intervals. In contrast, they use Venn-Abers calibration with the indicator outcome 1(Y ≤ t) to construct calibrated CDF estimates.
>
> 4. "The example presented in the experiment section is interesting ... "
>
>     **Response:** In the revised paper, we plan to include additional real data experiments, specifically the "bike," "bio,", "star", "concrete", and "community" datasets used in [1]. Our method generally will not fail to achieve self-calibration, barring failure of standard CP assumptions like exchangeability. However, there are scenarios where the prediction-conditional validity of our approach may lead to poorly adaptive prediction intervals relative to CP methods targeting feature-conditional validity.
>
>     In Appendix B.3, we use synthetic datasets to illustrate how prediction-conditional validity can approximate feature-conditional validity when the heteroscedasticity/variance of the outcomes is related to its conditional mean. In such cases, our approach offers narrow interval widths regardless of feature dimensions, leveraging model predictions as a scalar dimension reduction. We also show that in scenarios without a mean-variance relationship, our approach may provide poor feature-conditional coverage.
>
>     In the revised paper, we discuss these scenarios and point to the synthetic data experiments. To strengthen the real-data experiments, we will include the CQR method of [1] as a baseline for feature-conditional validity to investigate if prediction-conditional validity approximates feature-conditional validity in real data.
>
>
>    [1] Romano, Yaniv, Evan Patterson, and Emmanuel Candes. "Conformalized quantile regression." Advances in neural information processing systems 32 (2019).
>
>    [2] Gibbs, Isaac, John J. Cherian, and Emmanuel J. Candès. "Conformal prediction with conditional guarantees." arXiv preprint arXiv:2305.12616 (2023).
>
> 5. "Did your theoretical analysis offer any general insights..."
>
>     **Response:** Our main contribution is integrating two areas of trustworthy machine learning: (1) calibration of model outputs and (2) uncertainty quantification via prediction intervals, proposing self-calibration and self-calibrating conformal prediction. Our theoretical analysis offers further insights and potential extensions. Our techniques can analyze conformal prediction methods that involve calibrating model predictions followed by constructing conditionally valid prediction intervals. One could apply a feature-conditional CP method with conformity scores and/or the conditioning variables depending on calibrated model predictions and derive feature-conditional validity guarantees using a straightforward modification of our arguments. We plan to explore generalizations of our procedure in future work and have added a concluding paragraph discussing these insights and extensions.
>
> ## Limitations
>
> **Response:** We have moved our discussion of the limitations to its own subsection in Section 4. This paragraph includes all the limitations discussed in the checklist, as well as other limitations mentioned throughout the submitted version of the paper.

---

> > ### Comment · Reviewer_QroB · 2024-08-10
> >
> > Thank you for the detailed response. As most of my concerns are addressed, I'm happy to raise my score to 6.

---

### Official Review · Reviewer_bFVX · 2024-07-09

**Soundness:** 4
**Presentation:** 4
**Contribution:** 4
**Rating:** 7
**Confidence:** 4

**Summary:**

This paper introduces Self-Calibrating Conformal Prediction a novel uncertainty estimation framework combining Venn-abers calibration with conformal prediction. The output provides calibrated point predictions with associated prediction intervals with validity conditional on these model predictions.

**Strengths:**

1. Originality: The paper presents a novel combination of Venn-Abers calibration and conformal prediction, extending Venn-Abers to regression settings. This approach to achieving both calibrated point predictions and conditionally valid prediction intervals is novel.

2. Quality: The theoretical foundations are well-developed, with clear assumptions and complete proofs provided in the appendix. The authors prove important properties of their method, including perfect calibration of the Venn-Abers multi-prediction (Theorem 4.1) and self-calibration of the prediction interval (Theorem 4.2).

3. Clarity: The paper is well-structured and clearly written. The motivation and formulation are very well done.

4. Significance: The proposed method addresses an important ML uncertainty estimation problem - providing reliable uncertainty estimates that are both calibrated and conditionally valid. This problem is important in the context of trustworthy ML.

**Weaknesses:**

1. Limited experimental evaluation: While the MEPS dataset experiment is a nice case-study, the paper would benefit from evaluations on a wider range of datasets as is common in conformal prediction works. E.g. the widely used conformal regression datasets from CQR (https://arxiv.org/abs/1905.03222). The synthetic setups in the appendix help, but more real world datasets would be useful

2. Comparison to other conformal regression approaches: besides the baseline marginal CP, it would be helpful to see where other robust conformal approaches like CQR fit in compared to SC-CP— as they all should have good coverage and width, but it would be useful to see their calibration relative to the proposed method.

**Questions:**

1. Can the authors provide more insight into the trade-offs between computational complexity and accuracy when using the approximations suggested for non-discrete outcomes?

2. How sensitive is the method to the choice of the initial predictor f? Are there certain types of predictors that work particularly well or poorly with SC-CP? E.g. an xgboost was used how would the performance differ for other predictors f

**Limitations:**

The authors have done a good job of addressing the limitations of their work. They discuss potential computational challenges for non-discrete outcomes and suggest approximations.

Other limitations are briefly sprinkled throughout the paper. To improve this aspect, a suggestion is to include a dedicated "Limitations" section that consolidates and expands upon the current discussion of limitations scattered throughout the paper to make readers more aware of these factors.

---

> ### Author Rebuttal · Authors · 2024-08-07
>
> ## Response to Weaknesses
>
> 1. Thank you for this suggestions. In the revised paper, we will include additional real data experiments, specifically the "bike," "bio,", "star", "concrete", and "community" datasets used in CQR reference.
>
> 3. In the revised real experiments, we will include CQR as a feature-conditional baseline to investigate for which datasets prediction-conditional validity provides an adequate (or poor) approximation of feature-conditional validity.
>
> ## Response to Questions
>
> 1. Thank you for this question. In the revised version of the paper, rather than approximating the algorithm via discretization, we instead propose running the algorithm exactly using a discretized outcome and a discretized model. The outcome can be discretized by binning the outcome space into, say, 200 bins, and the model can be discretized similarly. In this case, the algorithm can be computed exactly, and the coverage and calibration guarantees are with respect to the discretized outcome. Discretizing the model output into 100-200 bins is generally sufficient to preserve predictive performance, especially isotonic regression/calibration already bins the model output. Discretizing the outcome allows the user to directly control how much the discretized outcome can deviate from the true outcome. We also discuss how to increase the width of the prediction interval by the outcome approximation error to guarantee coverage for the true outcome.▍
>
> 2. We have added a discussion on model choice to the problem setup and experiment section. Our method, like most conformal prediction methods, provides tighter prediction intervals as the model's predictiveness (e.g., MSE) improves. In our algorithm, isotonic regression learns an optimal monotone transformation of the original predictor (Theorem 4.3) and, therefore, can asymptotically only improve the MSE of the original predictor. However, if the original predictor is poorly predictive, the calibrated predictor, albeit calibrated, will typically also not be predictive. In addition, the usefulness of prediction-conditional validity may depend on the predictiveness of the model. An example where prediction-conditional validity is not useful is when the predictor is constant, in which case prediction-conditional validity reduces to marginal validity.▍
>
> ## Response to Limitations.
>
> 1. We have moved our discussion of the limitations to its own subsection in Section 4. This paragraph includes all the limitations discussed in the checklist, as well as other limitations mentioned throughout the submitted version of the paper.

---

> > ### Comment · Reviewer_bFVX · 2024-08-08
> >
> > Dear Authors
> >
> > Thank you for your response and the promised changes.
> >
> > My overall assessment of the paper is still positive, but it would have been helpful if the authors had attempted to tangibly show any of these promised changes — especially the additional datasets & CQR even with only a few seeds, as these are quite cheap to run and uploaded it in the response pdf.
> >
> > It would greatly help to be able to understand how these results would be positioned in and affect the updated paper.

---

### Decision · Program_Chairs · 2024-09-25

**Decision:**

Accept (poster)

**Comment:**

This paper combines conformal prediction with a regression extension of Venn-Abers calibration to provide calibrated point predictions and prediction intervals with finite-sample guarantees. Overall, the paper is written clearly, self-contained, and well-motivated. The extension of Venn-Abers calibration to regression settings is a novel and significant contribution. The empirical evaluation in the submission is somewhat lacking, focusing only on a single case study, and there are some clarity issues around the motivation for their approximation of feature-level conditioning.

While the authors promised additional experiments and clarifications, they did not include new experimental results during the rebuttal period. I am recommending that this paper is accepted despite these omissions, as I believe it will be of interest to the NeurIPS community. However, I urge the authors to put careful effort into the camera ready to deliver the promised additional results and address clarifications that came up during the reviewing process.